# COFLEX: A novel set point optimiser and feedforward-feedback control scheme for large flexible wind turbines

Guido Lazzerini[1], Jacob Deleuran Grunnet[2], Tobias Gybel Hovgaard[2], Fabio Caponetti[2], Vasu Datta Madireddi[2], Delphine De Tavernier[3], and Sebastiaan Paul Mulders[1]

[1]Delft Center for Systems and Control, Faculty of Mechanical Engineering, Delft University of Technology, Delft, the Netherlands
[2]Shanghai Electric Wind Power Generation European Innovation Center, Aarhus, Denmark
[3]Department of Flow Physics and Technology, Faculty of Aerospace Engineering, Delft University of Technology, Delft, the Netherlands

**Correspondence:** Guido Lazzerini (g.lazzerini@tudelft.nl)

**Abstract.** Large-scale wind turbines offer higher power output but present design challenges as increased blade flexibility affects aerodynamic performance and loading under varying conditions. Although flexible structures are considered in terms of (periodic) load control and aerodynamic stability, the impact of flexibility on the aerodynamic response of the blades is currently not fully addressed in conventional control strategies. The current state-of-the-art control strategy is the tip-speed ratio tracking scheme, which aims to maximise power production in the partial load region by maintaining a constant ratio between blade velocity and wind speed. However, this approach fails under large deformations, where the deflection and structural twist of the blade impact aerodynamic performance. This work aims to redefine the state-of-the-art wind turbine control with COFLEX (COntrol scheme for FLEXible wind turbines): a novel feedforward-feedback control scheme that leverages optimal operational set points computed by COFLEXOpt – a set point optimiser considering the effects of blade deformations on the aerodynamic performance and turbine loading. The proposed combined strategy consists of two key modules. The first module, COFLEXOpt, is an optimisation framework that provides controller set points while allowing constraints to be imposed on various operational, structural, and load properties, such as blade deflection and other structural loads. Set points obtained using COFLEXOpt are agnostic to operating regions, meaning that the operating region boundaries are optimised rather than prescribed. The second module is a feedforward-feedback controller and uses the set point mappings generated with COFLEXOpt, scheduled on wind speed estimates, to evaluate feedforward inputs and feedback to correct modelling inaccuracies and ensure closed-loop stability. A set point smoothing technique enables smooth transitions from partial to full load operations. The IEA 15 MW turbine is used as an exemplary case to show the effectiveness of COFLEX in maximising rotor aerodynamic efficiency while imposing blade out-of-plane tip displacement constraints. An analysis of the steady state optimisation results shows that accounting for blade flexibility leads to variable optimal tip-speed ratio operating points in the partial load region, and the collective pitch angle can be used to counteract blade torsion, maximising power coefficient while complying with imposed constraints. The established controller, tailored to track these optimised set points and operating points, was evaluated through time-marching mid-fidelity HAWC2 simulations across the entire operational range of the IEA 15 MW RWT turbine. These simulations, performed under uniform and turbulent wind inflows, demonstrate

excellent agreement between optimised steady states and median values obtained from HAWC2 simulations. Furthermore, the
generator power shows an increase of up to 5% in the partial load region compared to the reference scheme while maintaining
blade deflection at a similar level.

## 1 Introduction

While the European and international renewable energy targets for 2030 and 2050 provide an important framework for the
future development of wind energy (IEA, 2023), the drive for larger, multi-MW wind turbines is primarily motivated by the
ongoing efforts to reduce the cost of energy. This push is especially pronounced in the offshore wind sector, where the high costs
associated with installation favour the selection of larger turbines (Liang et al., 2021). However, enlarging components while
simultaneously aiming to keep costs low presents a significant challenge for wind turbine designers and manufacturers (Ja-
nipour, 2023). Cost-effective large structures become highly flexible, and turbines with higher power ratings are inherently
subject to higher loads (Sieros et al., 2012), coming from wind, inertia, and even sea waves in offshore installations (Veers
et al., 2023). These loads deform the structures, such as the turbine tower, but in particular, the blades. In contrast to stiffer,
smaller-scale turbines, blade flexibility heavily impacts aerodynamic and mechanical performance and results in complex sys-
tem dynamics (Pagamonci et al., 2023). Passive design techniques such as pre-coning, pre-bending, and bend-twist coupling
can mitigate some of these effects by modifying the geometrical and structural properties of the rotor. For instance, while
pre-coning and pre-bending can increase blade-to-tower clearance and increase the maximum swept area when the turbine is
operating at its rated condition, bend-twist coupling can be used to reduce aerodynamic loading passively (Sartori et al., 2018).
Nonetheless, these structural measures remain complementary to advanced active control, which can further optimise energy
capture and help decrease loads (Bortolotti et al., 2019). Extensive studies on aeroelastic interactions have led to structurally
feasible designs (Wang et al., 2016; Rinker et al., 2020; Escalera Mendoza et al., 2023) and the commercialisation of large
wind turbines with rated power reaching up to 20 MW (GE Renewable Energy, 2024; Vestas, 2024; Siemens Gamesa, 2024;
Memija, 2024) has demonstrated that scaling-up challenges can be successfully addressed. Concurrently, joint research teams
have designed bleeding-edge reference wind turbines (RWTs) for the wind energy community, pushing the rated power up to
22 MW (Zahle et al., 2024).

Conventional turbine controller designs drive the system to *optimal* operating points derived from steady-state calculations
which, in the partial load region, often assume an optimal constant tip-speed ratio and fixed collective pitch angle set point to
maximise power production (Hansen and Henriksen, 2013; Brandetti et al., 2023). An example of this approach is implemented
in the ROSCO controller, which employs tip-speed ratio tracking for generator torque control, aiming to maximise power
capture in the partial load region (Abbas et al., 2022). An even simpler approach is represented by the $K\omega^2$ controller, which
sets the generator torque in the partial load region proportional to the square root of the rotor speed via a constant gain $K$ (Pao
and Johnson, 2011). This approach, while still effective for present-day wind turbines (Brandetti et al., 2023), is also limited
by its dependence on an assumed power coefficient curve.

In fact, flexible blades of large wind turbines are subjected to heavy loads and undergo significant deformations, causing blade sections to deflect and twist from their unloaded positions (Trigaux et al., 2024). These structural changes alter the relative angle of attack experienced by the individual blade sections, which in turn affects the aerodynamic performance of the rotor.

While TSR-tracking and $K\omega^2$ control schemes have been successful in research and industrial turbines over the past decades, our work proposes a novel and combined set point optimisation and feedforward-feedback controller strategy to address the increased structural flexibility of next-generation turbines. Since the tip-speed ratio is the ratio between the blade tip speed and incoming wind speed, the same value for the tip-speed ratio can result from different combinations of wind speed and rotational speed, each producing different loading conditions. Hence, the effects of deformations are not captured when performance is parameterised solely by this quantity. Therefore, when considering the performance of larger, more flexible wind turbines, constant tip-speed ratio-based control strategies should be reconsidered. Instead, rotor and wind speed, which compose the tip-speed ratio, should be treated as independent variables in future control strategies. Moving away from constant tip-speed ratio assumptions allows us to explore control in a three-dimensional space, where rotor speed, wind speed, and collective pitch angle are considered independently to better account for flexible behaviour of the turbine.

To determine the optimised set points schedules, this work follows the emerging trend of calculating operating points by formulating the definition of steady-state set points as a nonlinear optimisation problem, with rotational speed and collective pitch angle as decision variables scheduled on wind speed (Pusch et al., 2023). Another example of implementing a variable steady-state schedule for the collective pitch angle in the partial load region was demonstrated in the recently published IEA 22 MW RWT design report (Zahle et al., 2024). In the IEA 22 MW RWT, the controller adjusts collective pitch angle set points in the partial load region with the two-fold objective of maximising the aerodynamic performance of the blades and ensuring peak-shaving of thrust. However, the concept of a variable optimal tip-speed ratio is not addressed, and details of the framework used to calculate the schedules have yet to be disclosed. An earlier example of deriving schedules for the steady-state operating points was provided by Bottasso et al. (2012). Set points were optimised for a representative 3 MW turbine to constrain the blade tip speed in the near-rated region, and an LQR controller was employed to perform power tracking. More recently, Petrović and Bottasso (2017) demonstrated that an optimal control employing online updates of power reference set points could be designed on top of a conventional controller to alleviate loads.

In this work, we advance state-of-the-art control for large-scale wind turbines by introducing the COFLEX scheme, which optimises turbine performance across the entire operational range while accounting for blade flexibility. Unlike conventional approaches that assume a unique *optimal* tip-speed ratio and fixed collective pitch angle, COFLEX leverages a set point optimisation framework (COFLEXOpt) to calculate schedules for the desired rotor speed and collective pitch for every wind speed, according to a constrained optimisation problem. This approach eliminates the need to predefine operating points for transitions between partial and full load regions, as the boundary between regions is optimised rather than fixed. Furthermore, COFLEXOpt enables the formulation of a constrained optimisation problem, allowing for the inclusion of specific constraints on various quantities, such as blade deflection and other structural properties.

Finally, we developed a feedforward-feedback controller to track the optimised set points. We based our performance calculations on representations of performance in a three-dimensional space where the rotational speed, the wind speed and the collective pitch angle are the independent variables. The implications and efficacy of the COFLEX scheme are demonstrated on the highly flexible IEA 15 MW RWT (Gaertner et al., 2020), where it achieves improved rotor power capture compared to the baseline control strategy, while ensuring compliance with load and deflection limits to maintain structural integrity.

Thereby, the key novelties and contributions of this paper are:

- Providing a set point optimisation scheme called COFLEXOpt calculating set points over the complete turbine operating range using one optimisation problem, adhering to operational and structural load constraints, and without the need for explicit definition of the partial to full load transition point;

- Improving the accuracy of rotor-effective wind speed estimation by decomposing the dependency of power coefficient information from tip-speed ratio to rotor speed and wind speed;

- Proposing a feedforward-feedback controller using and tracking the COFLEXOpt optimised set points, satisfying and adhering to the constrained optimisation objective(s);

- Demonstrating the capabilities and performance advantages of the proposed COFLEX in a higher-fidelity simulation environment using realistic wind conditions;

- Sharing COFLEX in a publicly available and freely-accessible online repository (Lazzerini et al., 2024).

This paper is organised as follows. Section 2 presents an overview of COFLEX. Section 3 provides the flexible model calculations used in the controller and a comparison with rigid model calculations to highlight the effects of flexibility on performance and to understand the importance of considering flexibility in the control problem. Section 4 defines the set point optimisation framework, named COFLEXOpt, with results of steady-state calculations. Section 5 exposes the improvements to the
wind speed estimator scheme and the details of the novel controller scheme and Sect. 6 demonstrates the capabilities of the novel control scheme in step-response and realistic wind conditions through time-domain simulations. Finally, conclusions and possible future developments are outlined in Sect. 7.

## 2   Overview of COFLEX

This section provides a comprehensive overview of the novel control scheme. We briefly introduce the key elements of the con-
trol architecture, describing how each component contributes to the overall scheme. Figure 1 offers a graphical representation of the paper's structure and main components of the control scheme.

    As seen in Fig. 1, we start by calculating the steady states of wind turbine operating points in a three-dimensional space. The steady-state operating points need to be expressed as functions in a three-dimensional space, in the form "$f(\omega, V, \beta)$", where $\omega$ is the rotational speed, $V$ is the wind speed and $\beta$ is the collective pitch angle.

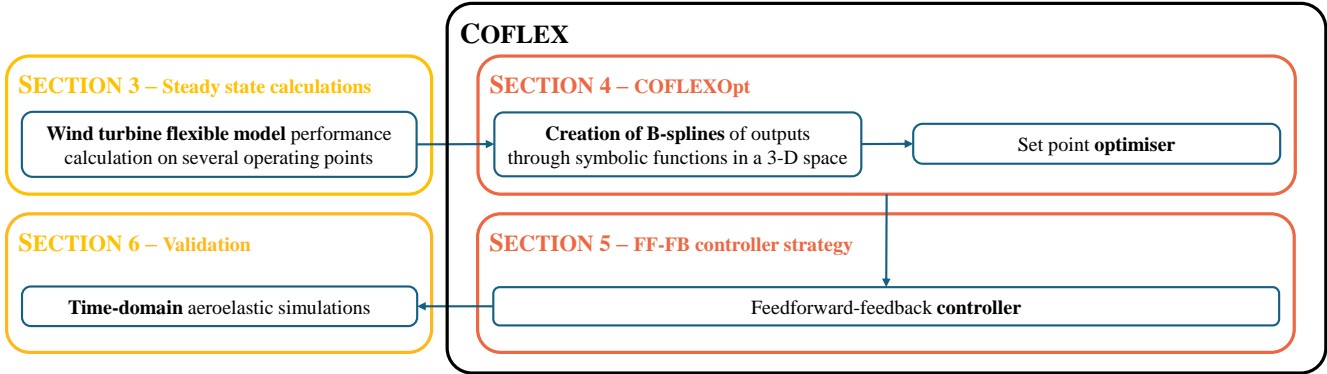

**Figure 1.** Schematic representation of the development of COFLEX, with indications of the main topics for each section of this manuscript. The steady-states calculation (Sect. 3) of performance is used as inputs to COFLEXOpt (Sect. 4). The optimised set points are tracked through a controller scheme (Sect. 5), which was validated with time-domain simulations (Sect. 6).

For this purpose, we use a flexible IEA 15MW RWT turbine model in HAWCStab2; this turbine was selected for its present-day relevance with modern turbines currently commercialised (GE Renewable Energy, 2024; Vestas, 2024) and because it is deemed to have a representative level of blade flexibility of such turbines. HAWCStab2 is an aeroelastic tool which solves the linearised dynamic equations of Blade Element Momentum Theory (BEMT) to calculate aerodynamic loads and implements an iterative process to account for deformed structures (Hansen, 2011). This tool was chosen for different reasons: First, it can take

into account large deformations of blades and structural couplings such as bend-twist in the loads calculations (Stäblein et al., 2017). Secondly, it provides a very fast computational time, which is crucial for evaluating performance across thousands of operating points that result from the combination of the three independent variables: rotational speed, wind speed and collective pitch angle, with sufficiently fine resolution. Hence, this tool offers a good trade-off between calculation accuracy and computational cost for operating-point evaluations.

Next, the post-processed performance data from the steady-state calculations serve as input to our set point optimisation framework. This framework operates within the Matlab-CasADi environment (Andersson et al., 2019), which implements the formulation and manipulation of symbolic functions and optimisation algorithms. Three-dimensional B-spline function interpolators of turbine performance metrics were used in the optimisation problem. COFLEXOpt is able to solve a constrained optimisation problem in the entire operating range of a wind turbine, providing optimised set points without prescribing oper-

ating regions. The constraints can be set to reflect design requirements.

     Once the set points are obtained by solving a numerical optimisation problem for the entire operating range of a wind turbine, they are used as inputs to the controller. We have developed a novel feedforward-feedback controller that utilises both generator torque and collective pitch angle to track the set points. The feedforward contributions, derived from COFLEXOpt mappings, are functions of the estimated wind speed. The feedforward control is implemented to accelerate the achievement

of the prescribed steady states such that the controller relies less on feedback to attain the desired operating point.

The feedback control component uses proportional-integral (PI) controllers to correct deviations from the optimised set points. A switching logic is implemented to allow for the alternate activation of the generator torque controller (active in the partial load region) and the collective pitch angle controller (active in the full load region) based on a set point smoothing technique adapted from the works of Schlipf (2021) and Zalkind et al. (2021). This technique forces the inactive controller to reach its saturation limit, preventing interference with the active controller and ensuring smooth operation under varying wind conditions.

A critical aspect of the feedforward control is its reliance on accurate wind speed estimation (Schlipf (2016) uses LiDAR measurements for feedforward control). In our work, the control system continuously estimates the wind speed to update the feedforward contributions accordingly. The wind speed estimator (WSE) used here is a modification of the immersion and invariance (II) estimator, first introduced in Ortega et al. (2011) and further developed in Liu et al. (2022b) and Brandetti et al. (2022).

We demonstrate the effectiveness of the control strategy on the IEA 15 MW RWT through a series of time-domain simulations carried out in the mid-fidelity aeroelastic code HAWC2 (Larsen and Hansen, 2007). These simulations, including both uniform wind steps and realistic turbulent wind conditions, demonstrate the control strategy's working principles, robustness, and efficacy in accurately tracking predefined operating points.

## 3 Effects of flexibility on steady-state performance of the IEA 15 MW RWT

In this section, we show how flexibility affects the steady-state power and thrust coefficients of the IEA 15 MW RWT. In Sect. 3.1, we will establish quantities to represent the wind turbine performance, which is the foundation for conventional controllers and - in an extended form - for the novel scheme. Section 3.2 presents the limitations of conventional tip-speed ratio tracking, which fails to account for structural deformations. Finally, we compare different performance metrics evaluated with rigid and flexible blade models, highlighting the significant performance variations induced by structural flexibility and the need for an optimised control scheme that incorporates these effects.

### 3.1 Fundamental wind turbine relations

First, we define the non-dimensional mechanical power coefficient as:

$$C_P = \frac{P}{\frac{1}{2}\rho V^3 \pi R^2},$$

(1)

where $P$ is the rotor mechanical power (W), $V$ is the rotor averaged wind speed ($\mathrm{m\,s^{-1}}$), $R$ is the blade radius (m) and $\rho$ is the air density ($\mathrm{kg\,m^{-3}}$). Note that we refer to the wind speed here as the spatial average of the longitudinal component of the atmospheric wind field at the rotor plane when unaffected by the presence of the wind turbine (see definition in Larsen and Hansen, 2007). In wind turbine design and analysis, non-dimensional parameters like $C_P$ are essential in evaluating wind turbine performance, as they provide a universal metric for comparing different turbines operating at various conditions.

We also introduce the torque coefficient as:

$$C_Q = \frac{Q}{\frac{1}{2}\rho V^2 \pi R^3}, \tag{2}$$

where $Q$ is the torque exerted on the rotor by the wind. The tip-speed ratio (TSR) is defined as the ratio between the tangential speed at the tip of the blade and the wind speed, as:

$$\lambda = \frac{\omega R}{V}, \tag{3}$$

calculated from the rotational speed of the rotor $\omega$. From Eqs. 1 and 2, we get the proportionality between the power and torque coefficients as follows:

$$C_P = \lambda C_Q. \tag{4}$$

Finally, we define the thrust coefficient to represent the force perpendicular to the rotor plane:

$$C_T = \frac{T}{\frac{1}{2}\rho V^2 \pi R^2}, \tag{5}$$

where $T$ is commonly known as thrust.

## 3.2 Decomposing the tip-speed ratio

In conventional controllers, the WSE and most gain-scheduling and peak-shaving routines are dependent and often calibrated using performance information where each entry is a function of $\lambda$ and the collective pitch angle $\beta$, i.e. functions in the form "$f(\lambda, \beta)$".

Moreover, optimal tip-speed ratio tracking control schemes are based on a constant $\lambda$ set point in the partial load region, which, to date, has been deemed to lead to optimal power extraction. The tip-speed ratio has effectively been used to define the aerodynamic state of reasonably rigid wind turbines. In fact, the aerodynamic performance is determined by the geometry of blade sections and angle of attack distribution, assuming Reynolds and Mach number variations are negligible (i.e., ignoring viscosity and compressibility effects on section aerodynamics). For a rigorous explanation, the reader is referred to the results of BEMT (Hansen, 2010).

However, when loads deform the blade shape and, consequently, the geometry of the sections, aerodynamic performance is altered. This variation is not captured by the tip-speed ratio alone because the same value of the tip-speed ratio may correspond to different combinations of $V$ and $\omega$. Due to blade flexibility, the traditional use of tip-speed ratio to parameterise the performance of wind turbines becomes inadequate. Consequently, it is necessary to parameterise the aerodynamic performance coefficients using three arguments, i.e. decomposing $\lambda$ into its components $\omega$ and $V$.

Two discrepancies with the actual aeroelastic behaviour of wind turbines arise when using aerodynamic performance coefficients parameterised on the tip-speed ratio for the design of conventional controllers:

1. The tools used for performance calculations may not account for structural flexibility primarily in the form of blade deformations, neglecting the effects of such deformations on aerodynamic behaviour, as already noted in Abbas et al. (2022).

2. The $C_P$ look-up tables are often calculated by fixing the wind speed to a reference value $V_{\text{ref}}$ representing the *average* or *rated* atmospheric condition for the turbine, and varying the rotational speed, resulting in a $C_P(\lambda,\beta)|_{V_{\text{ref}}}$ surface. This assumes that the wind turbine performance is unaffected by variations in loading and Reynolds number, which can change with different wind speeds. As suggested in Bottasso et al. (2012), a possible solution is to incorporate a third dimension when calculating $C_P$ look-up tables.

## 3.3 Aerodynamic performance evaluation using rigid and flexible blade models

To illustrate the discrepancies mentioned above, we compare $C_P$ and $C_T$ coefficients of the IEA 15 MW RWT using a rigid and flexible blade model. To balance computational effort and accuracy, the spacing in our grid is variable: it is refined in regions of particular interest—such as near the rated wind speed, where loads have a pronounced effect—and coarser in less critical regions. We then use HAWCStab2 to obtain the steady-state coefficients over a three-dimensional grid with 27 thousand operating points spanning various combinations of rotational speeds, wind speeds, and pitch angles. Specifically, the grid consists of:

- 20 rotor speeds $\omega$ (from 2 to 4 $\text{min}^{-1}$ in 1 $\text{min}^{-1}$ steps, from 5 to 9.5 $\text{min}^{-1}$ in 0.5 $\text{min}^{-1}$ increments, and from 10 to 16 $\text{min}^{-1}$ in 1 $\text{min}^{-1}$ steps),

- 30 wind speeds $V$ (from 2 to 7 $\text{m s}^{-1}$ in 1 $\text{m s}^{-1}$ steps, from 8 to 12.5 $\text{m s}^{-1}$ in 0.5 $\text{m s}^{-1}$ increments, and from 13 to 26 $\text{m s}^{-1}$ in 1 $\text{m s}^{-1}$ steps).

- 45 pitch angles $\beta$ (from $-5$ deg to 4.5 deg in 0.5 deg increments and from 6 deg to 30 deg in 1 deg increments).

No wind shear is considered here—i.e., we assume a spatially uniform inflow. This uniform inflow assumption arises from a limitation of HAWCStab2. In principle, it would be possible to incorporate wind shear by generating performance tables with a time-domain-based simulation tool such as HAWC2. However, creating such a large number of required operating points would be computationally infeasible.

The rigid model assumes infinitely stiff structures, while the flexible model considers fully flexible structures, where all linear and rotational deformation degrees of freedom are active. In the flexible model, each blade is divided into 20 sub-bodies using Timoshenko beam elements. These sub-bodies consist of two nodes with six degrees of freedom and coupled structural cross-sectional stiffness matrices (Stäblein et al., 2017), allowing for large deformations and modelling of bend-twist coupling (Lobitz and Veers, 2002). Table 1 provides an overview of the models and settings used in HAWCStab2.

Figures 2 and 3 show the power coefficient (a) and thrust coefficient (b) values, obtained fixing the pitch angle ($\beta = 0$ deg) and varying the wind speed (from 2 to 27 $\text{m s}^{-1}$) and rotational speed (from 2 to 16 $\text{min}^{-1}$) for a total of 600 combinations. The results are interpolated linearly on a finer grid for smoother variations in the analysed region. As expected from the discussion on tip-speed ratio, the rigid model (Fig. 2) exhibits $C_P$ and $C_T$ values that remain constant on constant tip-speed ratio lines. The results shown in Fig. 2 confirm that performance depends solely on the tip-speed ratio when using a purely aerodynamic solver (i.e. without the effects of deformations of blades) and neglecting Reynolds number variations along the blades.

**Table 1.** IEA-15MW RWT data and HAWCStab2 calculation settings. To obtain the HAWCStab2 **rigid** model from the original **flexible** model described in Gaertner et al. (2020), the elements of the stiffness matrices of the blades are increased by several orders of magnitude.

| IEA 15 MW key characteristics | | |
|---|---|---|
| Installation | Onshore | |
| Tower height | 130 m | |
| Hub height | 150 m | |
| Rotor diameter | 240 m | |
| Rotor and tower design | see Gaertner et al. (2020) | |
| **HAWCStab2 calculations settings.** | **Rigid** Model | **Flexible** model |
| Tower | Stiff Timoshenko beam | Flexible Timoshenko beam - 10 sections |
| Blades | Stiff Timoshenko beam | Flexible Timoshenko beam - 20 sections |

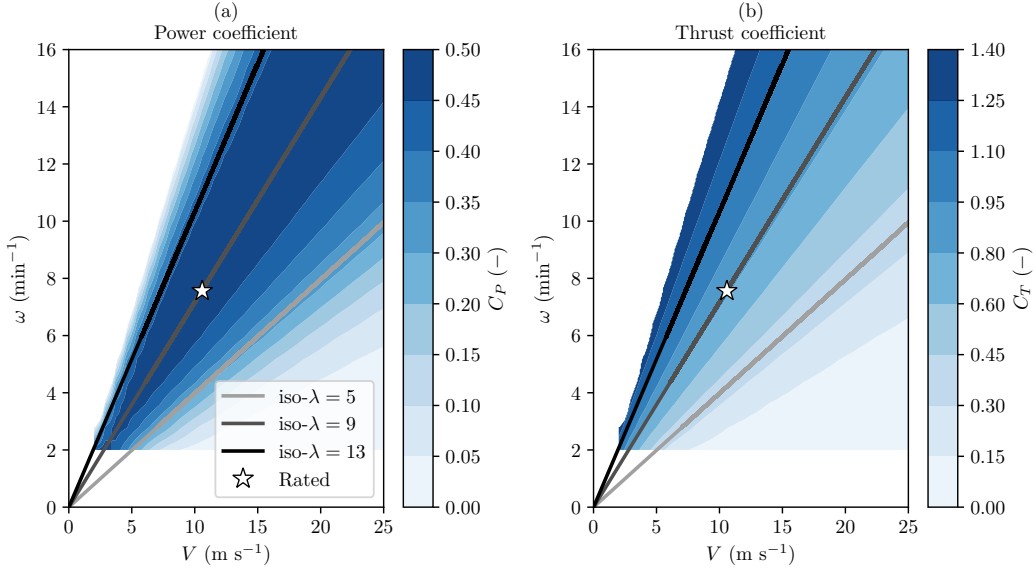

**Figure 2.** Power coefficient (a) and thrust coefficient (b) contour surfaces obtained varying rotational speed and wind speed for $\beta = 0$ deg in HAWCStab2 using the **rigid** model. Constant values can be found for both quantities along the iso-$\lambda$ lines (grey lines), indicating that for rigid blades and fixed collective pitch angle, the performance is *uniquely* dependent on $\lambda$. The rated operating point (white star) was obtained from Gaertner et al. (2020).

Figure 3 presents the power coefficient (a) and thrust coefficient (b) values obtained with the flexible model. In contrast to the
235 observations from the rigid model, values exhibit nonlinear, decreasing trends along iso-$\lambda$ lines. These differences stem from the coupled aerodynamic and structural response occurring in flexible blades: structural deformations introduce changes in the local angle of attack and in the relative wind velocity at the blade sections, causing deviations from the rigid model predictions.

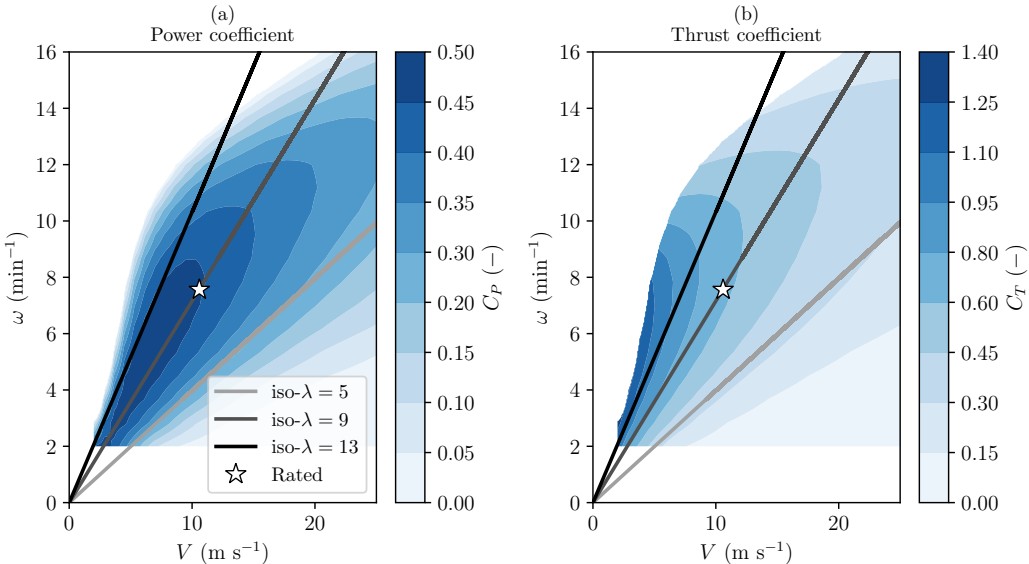

**Figure 3.** Power coefficient (a) and thrust coefficient (b) contour surfaces obtained by varying the rotational speed and wind speed for a constant collective pitch angle $\beta = 0$ deg in HAWCStab2 using the **flexible** model. Very different values can be found for both quantities along the iso-$\lambda$ lines (grey lines), indicating that for flexible blades and fixed collective pitch angle, the performance is dependent on the *exact combination* of $\omega$ and $V$. The rated operating point (white star) may not match the maximum power coefficient for this collective pitch angle configuration.

The varying trends along the iso-$\lambda$ lines in the flexible model highlight how flexibility-induced deformations impact both aerodynamic efficiency and loading. These effects become particularly pronounced at higher wind speeds and rotor speeds, where structural deformation is more significant. The trend is noticeable along all the iso-$\lambda$ lines, including the iso-$\lambda = 9$ line, which represents the partial load operational tip-speed ratio of the IEA 15 MW RWT.

In Fig. 4, the power coefficient (a) and thrust coefficient (b) are plotted against the wind speed, along iso-$\lambda = 9$ lines, for the rigid and the flexible models, to showcase the relative discrepancies for the same tip-speed ratio. Figure 4 shows that both $C_P|_{\lambda=9}$ and $C_T|_{\lambda=9}$ are constant when flexibility is neglected (blue line), whereas the fully flexible model (orange line) displays a substantial drop in power performance starting from $V = 7.5 \text{ m s}^{-1}$, and showing a more than 10% reduction in the $C_P$ with respect to the rigid model's corresponding value at $V = 10 \text{ m s}^{-1}$.

To illustrate how the structure of the blades changes, causing the reported reduction in power performance, two quantities representing structural deformations are shown: tip torsion in Fig.4 (c), indicating the structural twist of the blade tip section (positive when the structural twist decreases the angle of attack); and the out-of-plane (OoP) tip displacement in Fig.4 (d), which represents the distance of the blade tip section mid-chord point from the rotor plane, positive in the wind direction. From $V = 7.5 \text{ m s}^{-1}$ onwards, both these metrics exhibit significant differences compared to the values calculated with the rigid model.

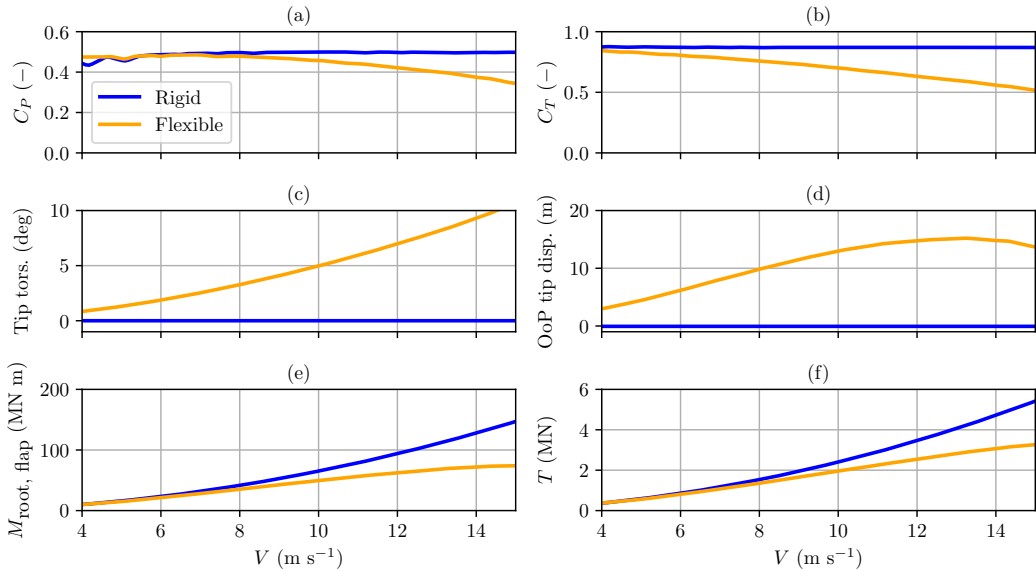

**Figure 4.** Comparison of performance, deformations and load characteristics of the two models at steady state, obtained varying wind speed and rotational speed, along a constant tip-speed ratio line, corresponding to the value $\lambda = 9$. All quantities were calculated using HAWCStab2.

The corresponding loads exerted on the rotor blades are shown in Fig.s 4 (e) and (f) in the form of the flap-wise bending moment at the root of the blades and the thrust force, respectively. When wind speed and rotational speed combinations produce rotor thrusts that exceed the peak value $T_{max} = 2750$ kN (as indicated in Gaertner et al. (2020)), the loads and deformations display highly nonlinear trends and influence one another. Under such conditions, large torsional deflections occur and, in turn, degrade performance while reducing loads. However, these operating points, corresponding to rotational speeds above $9$ min$^{-1}$ and wind speeds above $13$ m s$^{-1}$, lie well outside the normal steady-state operating conditions of the IEA 15 MW RWT. Consequently, these extreme deformations are not expected during typical turbine operation and are therefore considered unrealistic.

The findings in this section on the coupling between blade loading, structural flexibility, and the aerodynamic performance of the rotor suggest that constant tip-speed ratio tracking, a common control strategy for smaller and more rigid turbines, may no longer be sufficient to control large, flexible wind turbines optimally. These results indicate that there is room to optimise the power coefficient by accounting for blade flexibility early in the process of control design. They also show that flexible turbine calculations provide the opportunity for the incorporation of structural constraints once the set points are defined in a three-dimensional space of rotational speed ($\omega$), wind speed ($V$), and pitch angle ($\beta$). Based on the results and conclusions drawn in this section, we develop a new control scheme aimed at maximising energy capture while limiting excessive structural deformations.

The first model of the new scheme is a set point optimisation framework named COFLEXOpt, providing optimal (constrained) control set points and control inputs used to create steady-state mappings for the feedforward and feedback modules of the control scheme. The next section elaborates on the set point optimiser.

## 4 COFLEXOpt: Control set point optimiser

This section introduces the COFLEXOpt set point optimiser, which determines optimal operational points for large, flexible wind turbines. In Sect. 4.1, we formulate the optimisation problem for selecting set points based on turbine performance metrics and then explain the structure and implementation of the solver. Then, in Sect. 4.2 we show an illustrative example of the solution of the optimisation problem for two different wind speeds. Finally, in Sect. 4.3, we carry out set point optimisation for different control strategies.

### 4.1 Optimisation problem definition

Recently, Pusch et al. (2023) demonstrated a method for obtaining optimised operating points for wind turbines by solving an optimisation problem. In their study, steady-state set points were optimised by varying constraints, objective functions, and decision variables across different operating regions. As a consequence, the rated wind speed and operating regions were predefined. To simplify the optimisation setup and problem and possibly result in even more optimal solutions, our proposed framework solves the same optimisation problem to determine the set points, using a convex objective function over the entire operating range of a wind turbine – so in both partial and full load conditions. Notably, the decision variables in the optimisation problem remain unchanged over the entire range of operations. Hence, the subdivision of operating conditions into *regions* becomes irrelevant to the controller design. In addition, this optimiser allows for imposing constraints on various structural and operational quantities, such as thrust, blade deflection, tip-speed ratio, power output, and rotational speed, while still ensuring that an optimal solution is returned for each set of imposed constraints.

The general nonlinear optimisation problem can be written as follows:

$$
\begin{aligned}
&\min_{(\omega, \, \beta)} f_{\text{obj}}\left(\omega, \overline{V}, \beta\right) \ \forall \, \overline{V} \in [V_{\text{cut-in}}, \ V_{\text{cut-out}}] \,, \\
&\text{s.t. } \omega_{\min} \le \omega \le \omega_{\max} \ \& \ \beta_{\min} \le \beta \le \beta_{\max} \,, \\
&\qquad \mathbf{C}_{\text{iq}}\left(\omega, \overline{V}, \beta\right) \le 0 \,, \\
&\qquad \mathbf{C}_{\text{eq}}\left(\omega, \overline{V}, \beta\right) = 0 \,.
\end{aligned}
\tag{6}
$$

Where $\overline{V}$ represents a wind speed within the operating range $[V_{\text{cut-in}}, \ V_{\text{cut-out}}]$, $f_{\text{obj}}$ is a suitable objective function, $\omega_{\min}$, $\omega_{\max}$, $\beta_{\min}$, and $\beta_{\max}$ are box constraints on the decision variables, and $\mathbf{C}_{\text{iq}}$ and $\mathbf{C}_{\text{eq}}$ are in-equality and equality vector constraints, respectively. The decision variables are the rotational speed $\omega$ and the collective pitch angle $\beta$, as the performance of the wind turbine, including flexibility effects, can be expressed as functions of these variables and the wind speed. The versatility of this framework lies in the wide range of possible definitions for $f_{\text{obj}}$, $\mathbf{C}_{\text{eq}}$, and $\mathbf{C}_{\text{iq}}$. In particular, $\mathbf{C}_{\text{eq}}$, and $\mathbf{C}_{\text{iq}}$ can include any metrics representable in the $(\omega, V, \beta)$ space. Since the tip-speed ratio is decomposed into two separate variables, one can incorporate

non-linear constraints dependent on actual operating conditions. Examples include structural deflections, peak thrust (as in peak-shaving strategies), load-alleviation targets (e.g. bounding the root flapwise bending moment), or blade-span-dependent quantities (e.g. limiting angle of attack or relative velocities). Regarding the objective function $f_{\text{obj}}$, its formulation must yield unique and optimal solutions $(\omega^*, \beta^*)$ across the entire operating range. The primary objective is to maximise power capture (i.e. the power coefficient). The power output will also naturally be subject to an inequality constraint, ensuring the rated power is not exceeded. However, once the rated power limit is reached in the full-load region (i.e. $\overline{V} > V_{\text{rated}}$), infinitely many $(\omega^*, \overline{V}, \beta^*)$ combinations yield the power coefficient to produce the rated power and the maximisation of the power coefficient is not sufficient to produce unique solutions.

To address this, we introduce a secondary term in the objective function, resolving the non-uniqueness of the solution. This technique, also suggested in Iori et al. (2022), selects one point along the power coefficient iso-lines based on the minimisation of a secondary term in the objective function, resolving the non-uniqueness of the solution. In particular, this secondary term can have physical meaning: for example, if one selects the thrust coefficient, an increase in rotor loading is penalised in the optimal solution. Alternatively, one can penalise the torque coefficient, which ensures that the optimizer seeks the solution that yields the lowest rotor torque within the feasible region—helping to mitigate drivetrain loading. If the weight on this secondary term is kept sufficiently small, it effectively acts as a regularisation term while still retaining power maximisation as the primary objective. In our case, having defined custom inequality constraints that can include loads and structural deformations, we can directly target load alleviation through the imposition of limit (steady-state) values. As a result, we include only a small regularisation term in the objective function to ensure a limited impact on partial-load solutions. Hence, we propose maximising the power coefficient with a penalisation on the rotor torque coefficient for each wind speed $\overline{V}$, as follows:

$$f_{\text{obj}}\left(\omega, \overline{V}, \beta\right) = \left[-C_P\left(\omega, \overline{V}, \beta\right) + w_1 C_Q\left(\omega, \overline{V}, \beta\right)\right]. \tag{7}$$

In which the first objective has a unity weight, and the selection of $w_1$ remains the only tuning variable. Tuning parameters such as the weight $w_1$ is not a straightforward task. A similar challenge is reported in the work of Hovgaard et al. (2014), where multiple tuning parameters were required to balance competing goals in the objective function of a model predictive control scheme for wind turbines. This highlights the difficulty in tuning such parameters, which often involves trial and error to achieve the desired system behaviour. In our case, the torque term regularises the objective function in the full load region. In the selection of $w_1$, we should consider that increasing $w_1$ decreases the power coefficient in the partial load region and is therefore chosen small. In the remainder of this work, we set $w_1 = 0.01$. Because the power coefficient surface is relatively flat around its maximum in partial-load conditions, this small weighting factor has a negligible impact on the optimal set points in that region. However, it is sufficient to ensure unique solutions in the full-load region by regularising the objective function.

The formulation and solution of the optimisation problem in the set point optimiser framework is shown in Fig. 5. This figure illustrates the sequential steps in the optimisation process, starting from the initial calculation of performance metrics on a three-dimensional grid (upper left block), followed by the generation of multi-variate B-splines to ensure smooth, continuous performance functions (upper right block). These interpolated functions are then used to formulate the nonlinear Programming (NLP) problem (lower left block), which incorporates design constraints. The process concludes with the solution of this NLP,

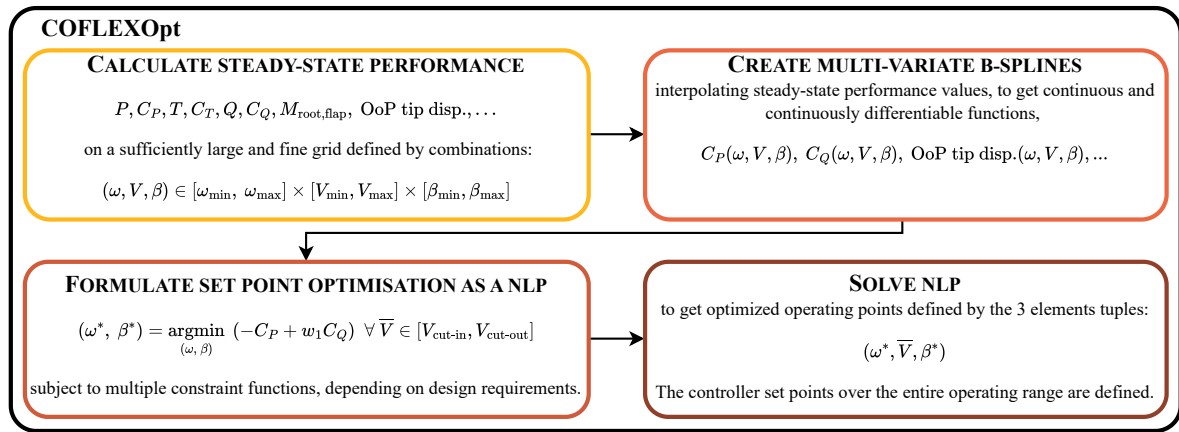

**Figure 5.** Block diagram of COFLEXOpt. The framework begins with the calculation of steady-state wind turbine performance over a large and fine grid of operating points defined by combinations of rotational speed, wind speed, and collective pitch angle. These performance values are interpolated using multi-variate B-splines to create continuous and differentiable functions, which are then used in the NLP optimisation process. The NLP is solved for each wind speed under the imposed objective function and constraints and defines the control set points across the entire operating range of the wind turbine.

yielding optimised operating points for the entire turbine operating range (lower right block). We implement the NonLinear Programming (NLP) process defined in Eq. (6), using CasADi and solve it with the IPOPT nonlinear solver (Wächter and Biegler, 2005). The most general optimisation problem for determining set points across the entire operating range of a wind turbine is defined as in Eq. 6 with the objective function of Eq. 7, with the following constraints:


$$
P_{\mathrm{g}}\left(\omega,\overline{V},\beta\right) \leq P_{\mathrm{g,rated}}\,,
$$
$$
Q_{\mathrm{g}}\left(\omega,\overline{V},\beta\right) \leq Q_{\mathrm{g,max}}\,,
\tag{8}
$$

where $P_{\mathrm{g,rated}}$ is the rated power and $Q_{\mathrm{g,max}}$ is the maximum admissible generator torque. The solution to this problem is represented by combinations of optimal values $(\omega^*, \beta^*)$ for each wind speed $\overline{V}$, gathered in set point mappings, which are then used in the feedback component of the controller.

## 4.2 Illustrative example: Optimisation working principles

Figure 6 illustrates the results of the optimisation process for two specific wind speeds, $10\ \mathrm{m\,s^{-1}}$ and $11\ \mathrm{m\,s^{-1}}$, chosen to represent partial and full load operating conditions. This figure visually demonstrates how the optimiser finds the best operating points while satisfying the required constraints in the different operating regions of a wind turbine, as we remark that COFLEXOpt is agnostic to regions. In each case, the power coefficient and torque coefficient are plotted against rotational speed and collective pitch angle. Optimal solutions found by COFLEXOpt are represented with red stars.

The grey regions in each plot represent infeasible zones where one or more constraints are violated, such as limits on power or torque. These areas indicate combinations of $\beta$ and $\omega$ that the optimiser cannot select, helping to emphasise the feasible

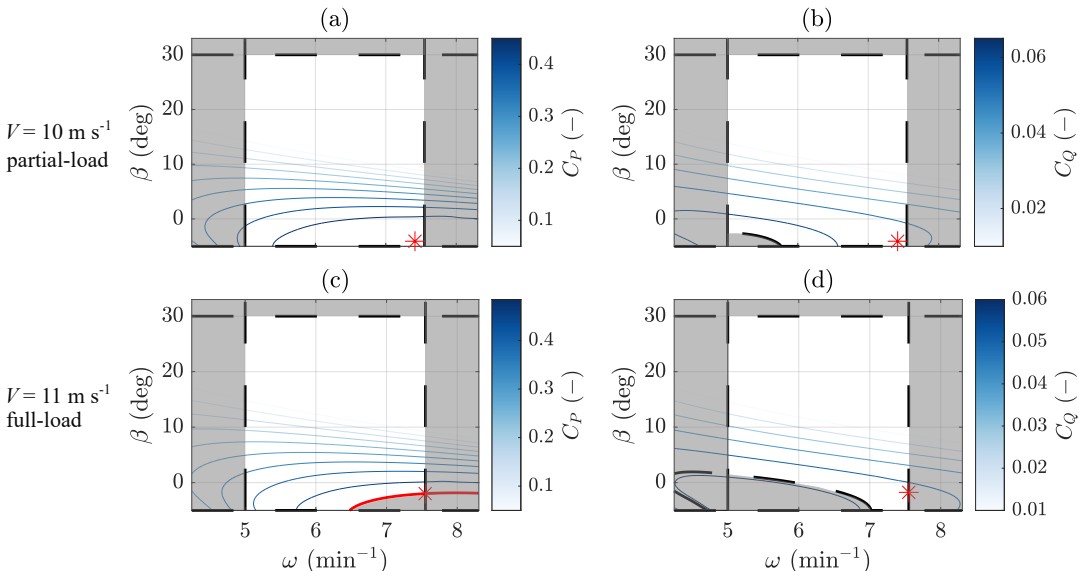

**Figure 6.** Visualisation of the solutions obtained from the NLP optimisation described by Eq. (8) for $\overline{V} = 10$ m s$^{-1}$ and $\overline{V} = 11$ m s$^{-1}$. The plots show the power coefficient ($C_P$) and torque coefficient ($C_Q$) as functions of rotational speed ($\omega$) and collective pitch angle ($\beta$). The optimal solutions, $(\omega^*, \beta^*)$, are marked with red stars, indicating the points that maximise $C_P$ while satisfying the constraints. Shaded grey areas represent regions that are infeasible due to these constraints. In subplot (c), the feasible solution space is bounded by the red line, where $C_P$ equals the rated power coefficient.

solution space. The difference between plots (a) and (c) versus (b) and (d) lies in the objective for each wind speed condition. In partial load (10 m s$^{-1}$), the optimisation focuses on maximising the power coefficient, as seen in plot (a). However, in the full load region, the rated power constraint leads to an infinite number of $(\omega, \beta)$ combinations along the red line highlighted
in Fig. 6 (c). The objective function becomes strictly convex due to the *small* contribution given by the torque coefficient term, as demonstrated by the iso-contours in Fig. 6 (d). These figures highlight the different sensitivities of power and torque coefficients to variations in rotational speed and collective pitch angle, which is exploited to find unique solutions to the optimisation problem over the entire operating range of a wind turbine. The optimal solution, which minimises $C_Q$, is found at the upper boundary of the rotational speed $\omega_{\mathrm{max}}$. This is because, in the NLP solved in this work, constraints on rotational
speed and generator torque were chosen according to values from Gaertner et al. (2020) to avoid major differences with the baseline controller design. For other applications of COFLEXOpt, such as optimising set points during the preliminary design of a wind turbine, relaxing these constraints is possible without sacrificing convergence capabilities.

### 4.3  Constrained set point optimisation for the IEA 15 MW turbine

In this section, we use COFLEXOpt to calculate set points for four different strategies, as summarised in Table 2. The first
strategy corresponds to the *Reference* approach, adopted for the IEA 15 MW RWT (Gaertner et al., 2020), and commonly

**Table 2.** Summary of the set points optimisation strategies analysed in this work. The reference strategy is the conventional TSR-tracking scheme. The other strategies aim to maximise power production in the partial load region while complying with increasingly tighter constraints on the blade out-of-plane tip displacement.

| Strategy | Objective | $\lambda^*(\overline{V})$ in partial load | $\beta^*(\overline{V})$ in partial load | OoP tip disp.$_{\text{max}}$ |
|---|---|---|---|---|
| **Reference** | See Gaertner et al. (2020) | Fixed | Fixed | Free |
| **Case 1** | Optimise $C_P$ with no constraints | Free | Free | Free |
| **Case 2** | Constraint on blade tip disp. set to **Reference** maximum | Free | Free | 13.6 m |
| **Case 3** | Tighter constraint on blade tip disp. | Free | Free | 10.0 m |

referred to as *optimal TSR tracking*, where a fixed optimal TSR value $\lambda^*$ is prescribed for the partial load region. The collective pitch angle is set to a minimum of 0 degrees, and the following formulation of the optimisation problem is used to obtain rotational speed set points:

$$\min_{(\omega,\,\beta)} \left[ -C_P\left(\omega,\overline{V},\beta\right) + w_1 C_Q\left(\omega,\overline{V},\beta\right) \right] \ \forall\, \overline{V} \in \left[3\ \text{m s}^{-1},\ 25\ \text{m s}^{-1}\right],$$

$$\text{s.t. } 5\ \text{min}^{-1} \leq \omega \leq 7.55\ \text{min}^{-1} \quad \& \quad 0\ \text{deg} \leq \beta \leq 30\ \text{deg},$$

$$\lambda^* = 9 \ \forall\, \overline{V} \in \left[V_{\omega_{\min}},\ V_{\text{rated}}\right],$$

$$P_{\text{g}}\left(\omega,\overline{V},\beta\right) \leq 15\ \text{MW},$$

$$Q_{\text{g}}\left(\omega,\overline{V},\beta\right) \leq 21.1\ \text{MN m}.$$

$$(9)$$

Note that this *fine-pitch* optimisation is only able to increase the power coefficient when the tip-speed ratio is constrained by the minimum rotational speed in the partial load region, with positive collective pitch angles. Under these assumptions, this strategy is not able to compensate for the flexibility effects illustrated in the previous section.

*Case 1* optimises the power coefficient in the partial load region without using a prescribed optimal tip-speed ratio and with no structural design constraints. The minimum collective pitch angle is set to -5 degrees. *Case 2* and *Case 3* include

upper limits on the OoP tip displacement at 13.6 metres and 10 metres, respectively. The first value was chosen based on the maximum value observed in the reference strategy, while the tighter constraint was introduced to evaluate the performance of the framework. To our knowledge, no previous studies or proposed frameworks have the capabilities to constrain steady-state structural properties such as OoP blade tip deflection directly in the optimisation problem, and this presents a significant contribution to COFLEXOpt. This quantity is relevant for the design of flexible wind turbines due to the risk of tower strikes.

It showcases the implementation of a critical structural performance constraint in our set point optimiser (Wang et al., 2023).

Figure 7 shows the resulting optimised set points (rotational speed, collective pitch angle, tip-speed ratio) and the corresponding steady-states for generator torque, blade OoP tip displacement and generator power for the four different strategies. While in the cut-in and full load region, the TSR values match for all cases (third plot), we observe that the optimal $\lambda$ from COFLEXOpt varies across the partial load region for all three cases. This again demonstrates that variable-$\lambda$ regulations lead to

improved performance, a tendency that is expected to intensify for more flexible rotors. The operating points of the collective

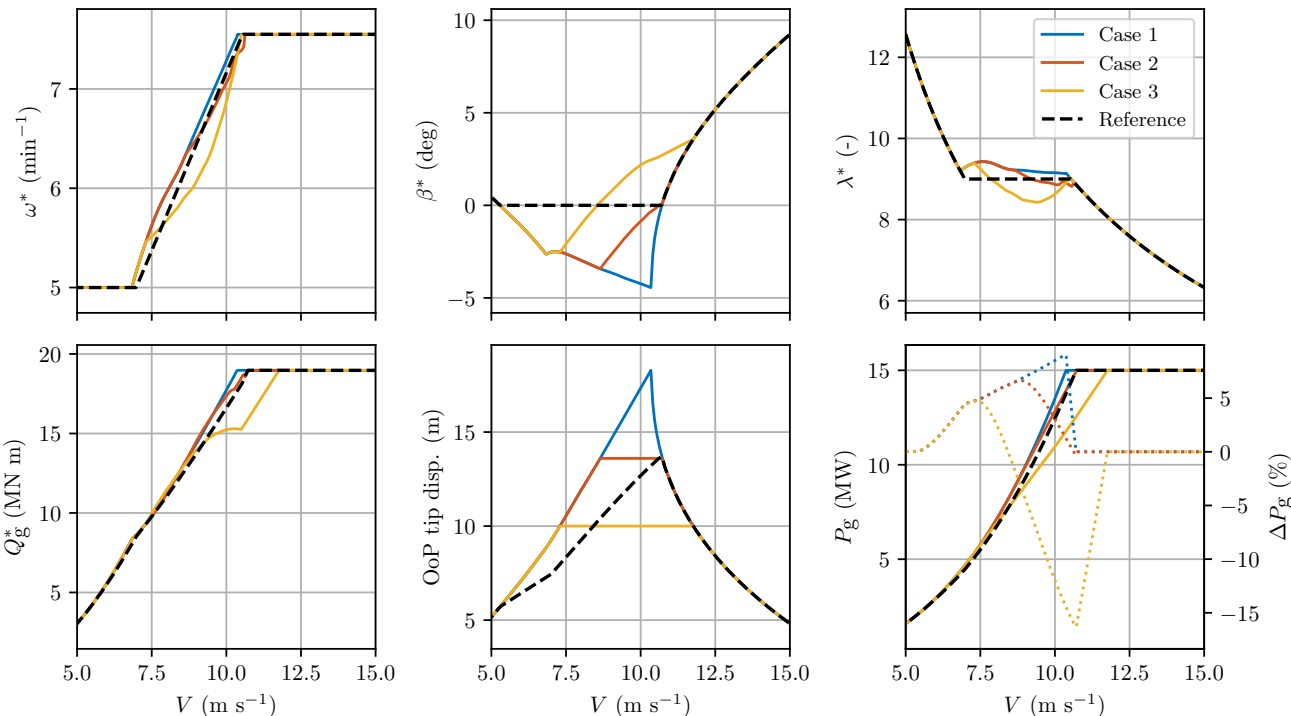

**Figure 7.** Comparison of optimised operating points for rotational speed ($\omega$), collective pitch angle ($\beta$), tip-speed ratio ($\lambda$), generator torque ($Q_g$), power output ($P_g$), and out-of-plane tip displacement, as obtained through the COFLEXOpt framework for different strategies (see Table 2). The plots cover wind speeds ranging from 5 to 15 metres per second. Each strategy reflects different optimisation priorities, with *Case 1* focusing on maximising power output without constraints, *Case 2* imposing a constraint on OoP tip displacement to match deflection levels of the reference strategy, and *Case 3* imposing even more conservative load constraints. The percentage differences in power output are shown relative to the reference strategy, highlighting consistent improvements in power generation for *Case 1* and *Case 2*. *Case 2* is particularly interesting for achieving higher power output while maintaining similar blade deflection levels compared to the reference.

pitch angle for Cases 1, 2, and 3 deviate from the reference values up to rated conditions. The optimisation framework allows pitching to stall, counteracting the effects of structural torsion on the blade and increasing the power output in the partial load region, as shown in the generator power plot. A different trend is observed in the constrained strategies, where the blades pitch to feather to relieve thrust force and facilitate the decrease in OoP tip displacement. The wind turbine performance output with the current recalculated optimised operating points returns higher power, with gains up to 10% for Case 1 and a consequent decrement of the rated wind speed.

Interestingly, the rated wind speed of Case 1 and Case 3 assumes different values w.r.t. the reference one, a direct result of the optimisation problem and imposed constraints, and is not predefined. This shows the major capability of the framework to arrive at the optimal solution and sets a new standard for deriving operating strategies for flexible turbines.

We notice an interesting effect on the operating points when the OoP tip displacement limit is active in the partial load region. Unlike a fixed tip-speed ratio strategy, COFLEXOpt allows concurrent changes in the rotational speed and pitch angle to find the optimal compromise between reducing loads and maximising the power coefficient. In this case, our approach takes advantage of the different sensitivities of the power coefficient and thrust coefficient to variations in pitch and rotor speed. In Case 3, the OoP tip displacement is effectively constrained to 10 metres, though this leads to power losses compared to the reference strategy. To correctly track the set points of the optimised strategies obtained with COFLEXOpt, we introduce a novel control scheme in the next section.

## 5    Feedforward-feedback control strategy

In this section, we describe the COFLEX control scheme. The diagram in Figure 8 retraces the main components of the control strategy, consisting out of an improved wind speed estimator for flexible turbines, set point smoother, and combined feedforward-feedback tracking control strategy.

In Sect. 5.1 we show a methodology to estimate the wind speed. Section 5.2 describes the generator torque and collective pitch angle controllers, discussing how feedforward set points strategies listed in Table 2 are tracked and how feedback terms correct deviations. Finally, Sect. 5.3 introduces the set point smoothing technique that manages transitions between control regions.

### 5.1    Wind speed estimator

The Wind Speed Estimator (WSE) employed in this work is part of the torque balance estimator class (Østergaard et al., 2007; Ortega et al., 2011; Liu et al., 2022b). Under the assumptions of measurable generator torque and rotational speed and a known power coefficient performance of the turbine, an estimate of the aerodynamic torque (or also rotor torque $Q_\mathrm{r}$) is used to derive an estimate of the rotor effective wind speed. The scheme used in this work is from Liu et al. (2022b) and employs the dynamic balance of rotor and generator torque at the rotor shaft, with a feedback loop for providing the rotor effective wind speed. The WSE illustrated in the block diagram of Fig. 9 is structurally identical to those shown in Liu et al. (2022b) and Brandetti et al. (2022). We refer the reader to these works for the full derivation and details on this WSE.

As demonstrated by Brandetti et al. (2022), the accuracy of wind speed estimates at steady state depends largely on the uncertainty in the power coefficient table, which is usually a function of $\lambda$ and $\beta$. As already demonstrated in Sect. 3, more accurate results can be obtained by using a flexible aeroelastic solver and removing the fixed tip-speed ratio approach to obtain a three-dimensional function for the power coefficient table, particularly when considering large and flexible wind turbines such as the IEA 15 MW RWT. To increase the accuracy of the estimated rotor torque for large, flexible rotors, we implement a modification to the schemes found in the literature by using the newly obtained power coefficient tables $C_P(\omega, V, \beta)$. To the authors' knowledge, this is the first effort to improve the accuracy of a torque balance-based WSE with a three-dimensional

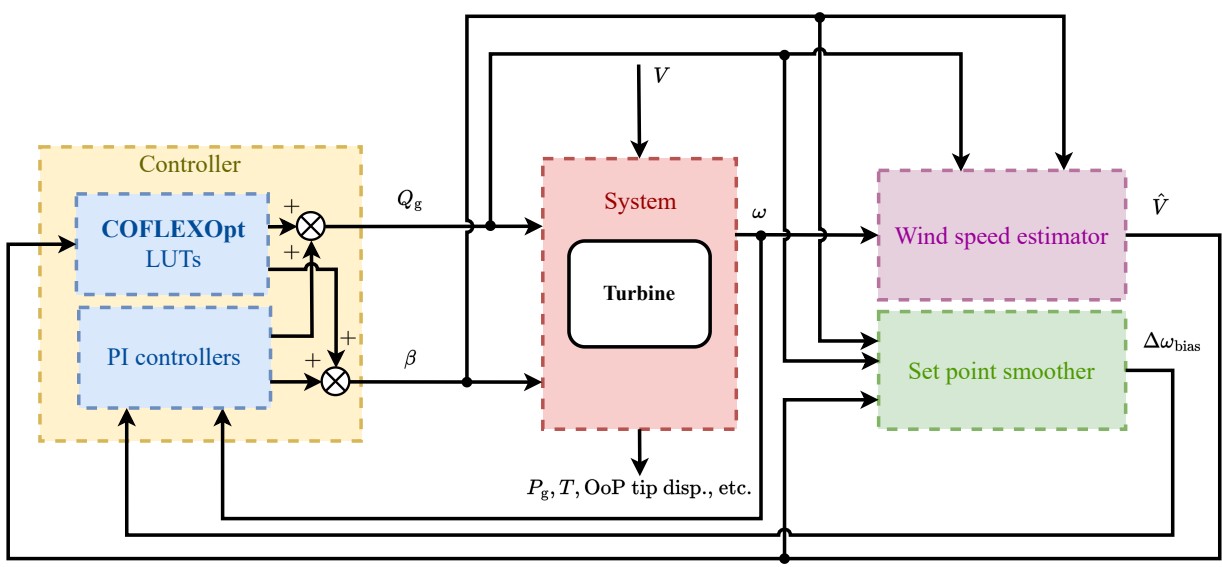

**Figure 8.** Block diagram of the control system architecture, illustrating the integration of the wind speed estimator, set point smoothing technique, and the feedforward-feedback controller. The diagram shows how the estimated wind speed ($\hat{V}$) is used in conjunction with COFLEXOpt look-up tables (LUTs) to determine the optimal set points for generator torque and collective pitch angle. The set point smoothing technique is employed to ensure smooth transitions between control modes, with the rotational speed set point bias ($\Delta\omega_{\text{bias}}$) being a key element in smoothing the control signals. This figure provides an overview of the components and their interactions within the control system.

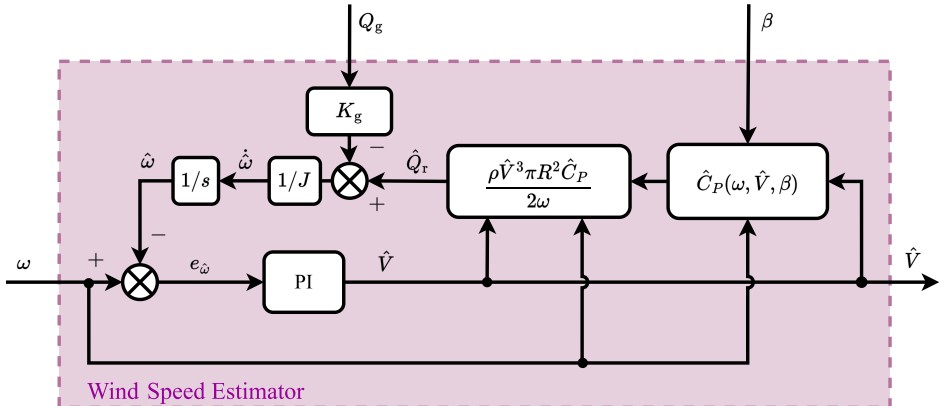

**Figure 9.** Detailed block diagram of the wind speed estimator (WSE) used in this study, based on modifications to the scheme presented by Brandetti et al. (2023). The WSE estimates the aerodynamic torque and wind speed by balancing rotor and generator torques, using a three-dimensional power coefficient table that accounts for blade flexibility.

**Table 3.** Summary of the Wind Speed Estimator (WSE) configurations analysed in this study, showing the model type, the parameterisation of the power coefficient table $C_P$, and the maximum steady-state error in estimated wind speed $e_{\hat{V}}$. The three cases include: (1) a rigid model using $C_P(\lambda,\beta)$ calculated at a reference wind speed; (2) a flexible model with $C_P(\lambda,\beta)$ calculated at the same reference wind speed; and (3) an enhanced flexible model with $C_P(\omega,V,\beta)$ to improve accuracy by capturing effects of rotational speed, wind speed, and pitch angle variations.

| WSE Case | Model | $C_P$ table | max $(|e_{\hat{V}}|)$ at steady state |
|---|---|---|---|
| **Rigid** | Rigid | $C_P(\lambda,\beta)|_{V=9 \text{ ms}^{-1}}$ | 3.5% |
| **Flex. 1** | Flexible | $C_P(\lambda,\beta)|_{V=9 \text{ ms}^{-1}}$ | 2.5% |
| **Flex. 2** | Flexible | $C_P(\omega,V,\beta)$ | 0.5% |

power coefficient table. Now, the system equations of the wind speed estimator are given as:

$$\begin{cases} \dot{\hat{\omega}} = \dfrac{\rho \hat{V}^3 \pi R^2 C_P(\omega,\hat{V},\beta)}{2J\omega} - \dfrac{K_{\text{g}}}{J}Q_{\text{g}}, \\ e_{\hat{\omega}} = \omega - \hat{\omega}, \\ \hat{V} = K_{\text{W,P}}e_{\hat{\omega}} + K_{\text{W,I}}\int e_{\hat{\omega}}(\tau)d\tau, \end{cases} \tag{10}$$

where a constant value $K_{\text{g}}$ represents the mechanical efficiency between the rotor and the generator, and $J$ is the total of the rotational inertia of the rotor, drivetrain, and generator shaft (recall that the IEA 15 MW RWT employs direct-drive technology). The estimated rotational acceleration $\dot{\hat{\omega}}$ is used to obtain an estimated rotational speed $\hat{\omega}$. A feedback loop with proportional

and integral gains ($K_{\text{W,P}}$ and $K_{\text{W,I}}$) is used to obtain an estimate of the wind speed $\hat{V}$.

To verify the improved performance of the WSE with an additional power coefficient table dimension, three time-domain simulations of the IEA 15 MW RWT were performed with uniform wind steps of $1 \text{ ms}^{-1}$ ranging from 3 to 11 $\text{ ms}^{-1}$, each step lasting 300 seconds. To analyse the accuracy of the steady-state wind speed estimation, we implemented a $K\omega^2$ scheme, selecting the gain $K$ according to the method in Pao and Johnson (2011). The constant $K$ was calculated based on the optimal

tip-speed ratio and corresponding maximum power coefficient prescribed by the IEA 15 MW RWT baseline design, reverting to the standard constant optimal tip-speed ratio assumption. In doing so, the steady-state behaviour is fully specified by the gain $K$ so that the generator torque controller does not rely on wind speed estimates. This approach decouples the steady-state performance of the WSE from other control routines, allowing us to evaluate the estimator without interference from the control tuning parameters. The $K\omega^2$ controller used in this section serves only as a convenient means to assess the WSE steady-state

performance. Three different schemes for the WSE were analysed, as summarised in Table 3: the first, named *Rigid*, being a WSE in which the $C_P$ table was calculated with a rigid model and parameterised on tip-speed ratio and collective pitch angle; in the *Flex. 1* WSE, the $C_P$ table was calculated taking into account flexibility and parameterised on tip-speed ratio and collective pitch angle; and in *Flex. 2*, the $C_P$ table was calculated with flexibility and parameterised on wind speed, rotational speed, and collective pitch angle. All $C_P$ tables were obtained using HAWCStab2 with the same models described in Sect. 3.

As shown in Table 3 and Fig. 10, the WSE with a three dimensional $C_P$ table *Flex. 2* outperforms the first two schemes in estimating the wind speed value in this operating region, with lower mean errors at steady state. As seen in Fig. 10, the *Rigid*

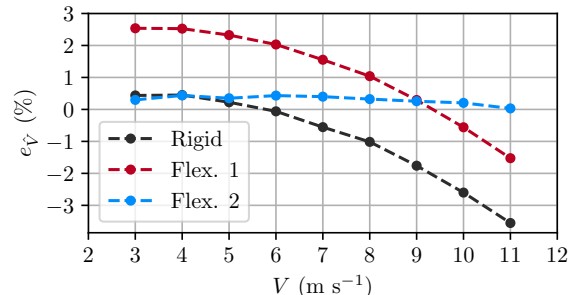

**Figure 10.** Evaluation of wind speed estimation accuracy with different WSE configurations. Percentage error in estimated wind speed ($e_{\hat{V}}$) as a function of actual wind speed during a simulation with uniform wind steps ranging from 3 to 11 metres per second. Data points represent the average of the final 100 seconds of each wind step after reaching steady state. The *Flex. 2* results, obtained using HAWC2 simulations for the IEA 15 MW RWT, demonstrate the improved accuracy of using the three-dimensional $C_P(\omega, \hat{V}, \beta)$ table to reduce estimation errors in the partial-load region.

WSE shows a small steady-state error for low wind speed cases up to 7 metres per second. *Flex. 1* is only able to estimate the wind speed with a small error in the neighbourhood of the wind speed which was chosen to calculate the $C_P(\lambda, \beta)$ table ($V = 9 \text{ m s}^{-1}$). The novel, improved scheme *Flex. 2* is able to estimate the wind speed at a steady state with a significantly

smaller error due to the improved match of the estimated rotor torque in the WSE model with the simulation model. The speed of convergence of the estimate to its steady-state value in all three cases analysed here is essentially related to the choice of the gains ($K_{\mathrm{W,P}}$ and $K_{\mathrm{W,I}}$). When the WSE is integrated into a controller scheme (i.e., $\hat{V}$ is used to compute inputs to the controller), tuning of the gains is needed as they become part of a dynamic feedback loop.

### 5.2 Generator torque and collective pitch angle controllers

The generator torque and collective pitch angle controller developed in this work implements feedforward set points parameterised on the wind speed estimate and, following well-established methodologies to control wind turbines (Bossanyi, 2003; Pao and Johnson, 2011), includes two PI feedback controllers to regulate the rotor speed. A set point smoothing technique allows switching between the two controllers by forcing the inactive controller to saturation. The scheme of Fig. 11 shows the controller implementation.

The following control laws are implemented for the generator torque ($Q_{\mathrm{g}}$) and collective pitch angle ($\beta$) command inputs to the system:

$$Q_{\mathrm{g}} = Q_{\mathrm{g, FF}}^{*} + \Delta Q_{\mathrm{g, FB}}, \tag{11}$$

$$\beta = \beta_{\mathrm{FF}}^{*} + \Delta \beta_{\mathrm{FB}}, \tag{12}$$

where the feedforward contributions $Q_{\mathrm{g, FF}}^{*}$ and $\beta_{\mathrm{FF}}^{*}$ are calculated with COFLEXOpt and are extracted from set point mappings

as a function of estimated wind speed $\hat{V}$ (as indicated by the functions $f(\hat{V})$ and $g(\hat{V})$ in Fig. 11). These quantities represent

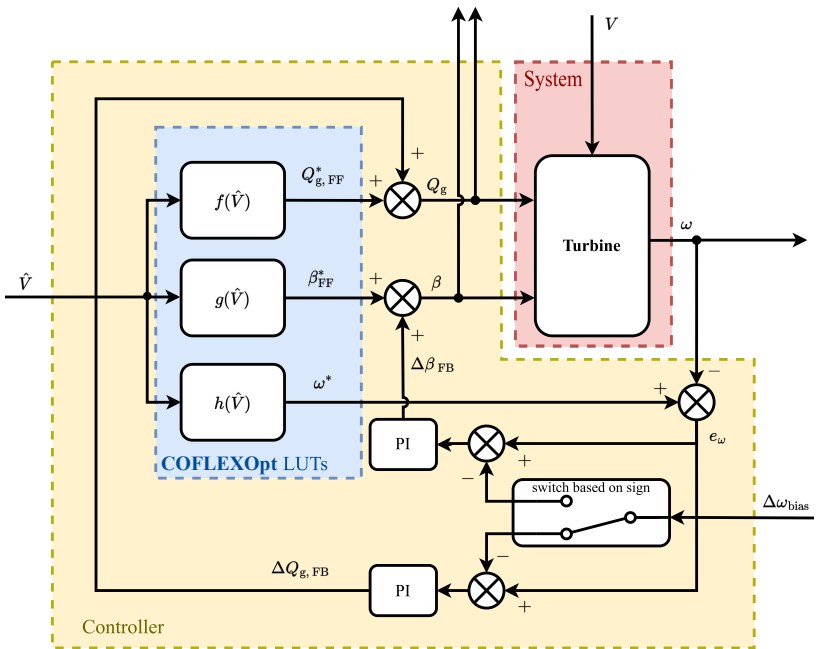

**Figure 11.** Block diagram detailing the implementation of the feedforward-feedback controller developed in this study. The controller uses feedforward set points derived from COFLEXOpt set point mappings and adjusts the generator torque and collective pitch angle outputs based on the estimated wind speed. Feedback contributions are calculated from the rotational speed error with proportional-integral (PI) controllers to improve stability and prevent model mismatch errors.

the desired steady-state set points for the entire operating range of the wind turbine and depend on the design requirements and optimisation strategy.

The feedback terms are calculated based on the rotational speed error, which is calculated as follows:

$$e_\omega = \omega^* - \omega. \tag{13}$$

Thus, the feedback contributions are given by:

$$\Delta Q_{\text{g, FB}} = K_{\text{P,Q}} e_\omega + K_{\text{I,Q}} \int e_\omega(\tau) d\tau, \tag{14}$$

$$\Delta \beta_{\text{FB}} = K_{\text{P},\beta} e_\omega + K_{\text{I},\beta} \int e_\omega(\tau) d\tau, \tag{15}$$

where the two gains for the generator torque contribution $K_{\text{P,Q}}$ and $K_{\text{I,Q}}$ must be defined so that $\Delta Q_{\text{g,FB}} < 0$ leads to acceleration of the rotor rotational speed when $e_\omega > 0$, while $K_{\text{P},\beta}$ and $K_{\text{I},\beta}$ must be defined so that $\Delta \beta_{\text{FB}}$ is negative (i.e., the
blades pitch towards stall, increasing the aerodynamic torque of the turbine) when $e_\omega > 0$. To satisfy controller performance requirements, such as overshoot and rise time, proper tuning of the gains $(K_{\text{P,Q}}, K_{\text{I,Q}}, K_{\text{P},\beta}, K_{\text{I},\beta})$ is necessary.

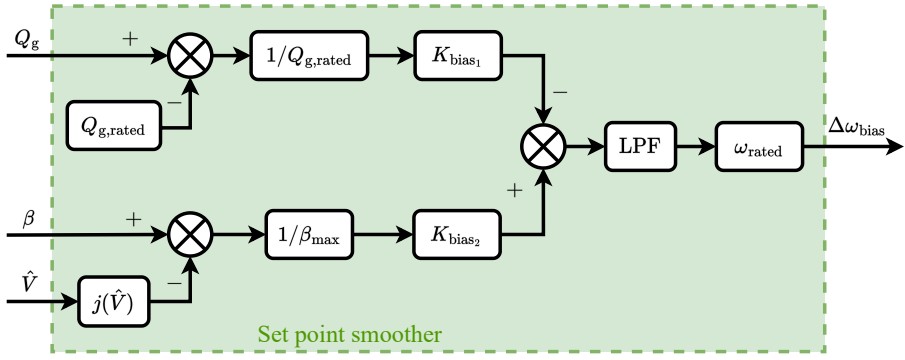

**Figure 12.** Schematic representation of the set point smoothing technique used to manage transitions between control regions. The technique applies a rotational speed set point bias to either the generator torque or collective pitch angle PI controller, depending on the operational region of the turbine. The smoothing function ensures that one controller is always saturated while the other is active.

At the same time, a bias $\Delta\omega_{\text{bias}}$ is introduced to the inactive controller set point through a switching logic. This technique for smoothing the set point in the switching region is implemented similarly as in Abbas et al. (2022) and explained in further detail in the following section.

### 475   5.3   Set point smoothing technique

A set point smoothing technique is described here to ensure a continuous transition between partial and full-load operations. To this aim, a bias is introduced in the reference set points of the two controllers. This set point bias is used to force one of the two PI controllers to saturate when the other is active. When the generator torque PI controller is active, i.e., in the partial-load region, the pitch controller should be forced to its lower saturation limit. Vice-versa, in the full-load region, the generator torque

should reach its upper saturation limit. The set point smoothing technique is represented by the block scheme of Fig. 12.

The following equations are used to calculate the contributions for the rotational speed set point bias $\Delta\omega_{\text{bias}}$:

$$\Delta\omega_{\text{bias}_1} = \frac{Q_{\text{g, rated}} - Q_{\text{g}}}{Q_{\text{g, rated}}}, \tag{16}$$

$$\Delta\omega_{\text{bias}_2} = \frac{\beta - j(\hat{V})}{\beta_{\text{max}}}, \tag{17}$$

where $Q_{\text{g, rated}}$ and $\beta_{\text{max}}$ represent the upper saturation limits of the generator torque and collective pitch angle, respectively.

In contrast, the function $j(\hat{V})$ represents the lower varying saturation limit for the collective pitch angle. We developed a new methodology to obtain $j(\hat{V})$. This function is introduced to ensure that the collective pitch angle correctly saturates to the prescribed set points in the partial-load region while preventing aerodynamically unstable behaviour in the full-load region and

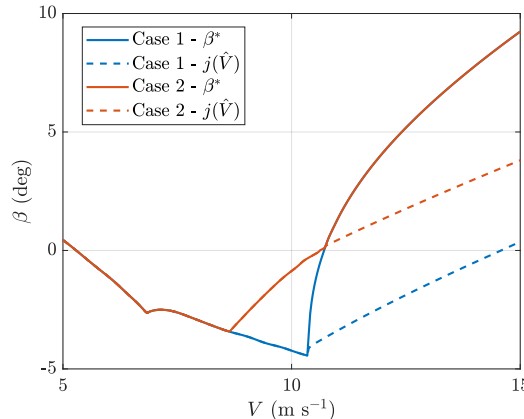

**Figure 13.** Collective pitch angle set points, and functions $j(\hat{V})$, representing the varying saturation limit. These functions ensure that the collective pitch angle correctly saturates to the prescribed set points in the partial-load region while maintaining stable operation in the full-load region. These functions were calculated for different set point optimisation strategies, and they are plotted here for *Case 1* and *Case 2*.

is obtained by solving a nonlinear program similar to Eq. (9):

$$
\begin{aligned}
(k(\overline{V}), j(\overline{V})) &= \underset{(\omega,\,\beta)}{\operatorname{argmin}} \left[ -C_P\left(\omega,\overline{V},\beta\right) + w_1 C_Q\left(\omega,\overline{V},\beta\right) \right] \; \forall\, \overline{V} \in \left[3 \text{ m s}^{-1},\, 25 \text{ m s}^{-1}\right], \\
&\text{s.t. } 5 \text{ min}^{-1} \le \omega \le 7.55 \text{ min}^{-1} \quad \& \quad -5 \text{ deg} \le \beta \le 30 \text{ deg}, \\
&\text{OoP tip disp.}\left(\omega,\overline{V},\beta\right) \le \text{OoP tip disp.}_{\text{max}},
\end{aligned}
\tag{18}
$$

without constraints on the maximum power and torque, but imposing the same maximum level on OoP tip displacement of the chosen set point strategy. A key motivation for deriving the lower pitch saturation limit from the "reduced" optimisation in Eq. 18 is to systematically obtain minimum pitch schedules that comply with the constraints imposed in COFLEXOpt optimised operating points and avoid stall. By defining an objective function that maximises aerodynamic efficiency (i.e. the power coefficient) and retaining the OoP tip displacement constraint, we ensure that at full load, the minimal-pitch operating

point (for any rotor speed–wind speed combination) remains above the stall onset value. This preserves aerodynamic stability and avoids stalled blades even if the turbine briefly operates at that minimal pitch. In contrast, simpler schedules (e.g., setting $j(\hat{V})$ to the pitch angle at rated conditions) may produce stalled conditions or violate tip-displacement limits for wind speeds in full-load operations.

     By solving this NLP, we obtain the two mappings $k(\overline{V})$ and $j(\overline{V})$ over the wind speed interval. The function $j(\overline{V})$ is used

to track the collective pitch angle set points in the partial-load region while being compliant with the design requirement and producing stable operating points for the wind turbine in the full-load region. This function was calculated for each set points strategy obtained with COFLEXOpt and is represented for illustrative purposes in Fig. 13 for Cases 1 and 2.

The contributions to the set point bias calculated in Eq. (16) and Eq. (17) are normalised and weighted so that the final value can be calculated as:

$$\Delta\omega_{\text{bias}} = \omega_{\text{rated}} \left( K_{\text{bias}_2} \Delta\omega_{\text{bias}_2} - K_{\text{bias}_1} \Delta\omega_{\text{bias}_1} \right), \tag{19}$$

In which the two gains $\{K_{\text{bias}_1}, K_{\text{bias}_2}\} \in \mathbb{R}^+$ are similar to the ones introduced in Abbas et al. (2022) and can be tuned to regulate the smoothness of the transition from one PI controller to the other. The signal $\Delta\omega_{\text{bias}}$ is also low-pass filtered to prevent high-frequency oscillations. In particular, we used a discrete-time first-order filter with a cut-off frequency of $0.2\pi \,\text{rad}\,\text{s}^{-1}$. The sign of this function depends on which one of the two controllers is saturated. In the partial-load region, $\Delta\omega_{\text{bias}_2} = 0$ and $\Delta\omega_{\text{bias}} < 0$, while if the generator torque is saturated, $\Delta\omega_{\text{bias}_1} = 0$ and $\Delta\omega_{\text{bias}} > 0$. A switching logic, which applies a bias to the two different set point inputs to the PI controllers, can be implemented based on the sign of $\Delta\omega_{\text{bias}}$:

$$\begin{cases} \text{if} & \Delta\omega_{\text{bias}} < 0 \rightarrow e'_\omega = e_\omega - \Delta\omega_{\text{bias}}, & \text{in collective pitch angle PI controller}, \\ \text{else} & \Delta\omega_{\text{bias}} \geq 0 \rightarrow e'_\omega = e_\omega - \Delta\omega_{\text{bias}}, & \text{in generator torque PI controller}. \end{cases}$$

In this way, in the partial-load region, the collective pitch angle controller receives a biased, higher set point $e'_\omega$, which pushes the blades to pitch to stall, forcing the controller to its lower saturation limit (i.e., the function $j(\hat{V})$). In the full-load region, the generator torque reaches its upper saturation limit $Q_{\text{g, rated}}$, and $\Delta\omega_{\text{bias}}$ changes sign, becoming positive. The switching logic applies a negative bias $(-|\Delta\omega_{\text{bias}}|)$ to the generator torque controller, which, in the attempt of trying to decelerate the rotor, is forced to its upper saturation limit, and the collective pitch angle controller becomes active. In the transition zone, the alternating activation of the controllers is smoothed by the presence of a low-pass filter in the rotational speed bias. To ensure a smooth transition, the gains $K_{\text{bias}_1}$ and $K_{\text{bias}_2}$ were re-tuned w.r.t. the values that can be found in Abbas et al. (2022).

## 5.4 Integration of WSE, controllers and set point smoother

The integration of the WSE, the set point smoothing technique, and the PI controllers leads to the novel control scheme for large, flexible wind turbines shown in Fig. 14.

This control scheme leverages feedforward action to achieve the desired set points, while feedback loops work to enhance stability, correct (tracking) errors, and add resiliency to disturbances and noise. However, its overall tracking performance is dependent on the accuracy of the internal power coefficient table. The wind speed estimation relies on this table, so any bias in the power coefficient data propagates into the estimates. As demonstrated by Brandetti et al. (2022), for the WSE-TSR tracking scheme, whenever the controller's reference is scheduled based on wind speed estimates, the system converges to a steady state that reflects this bias. In other words, the controller is capable of tracking a reference, but the reference itself is shifted from the true optimal operating point. This is essentially the same phenomenon encountered in standard tip-speed ratio tracking, where the *optimal* set point is also calculated offline using nominal aerodynamic data; if the real performance deviates from that nominal data, the turbine will no longer be operating at the true optimum. Our scheme will similarly be affected by inaccuracies in the internal power coefficient table, even though it maintains effective reference tracking. A potential mitigation of the bias introduced by modelling inaccuracies would be to schedule the feedforward input on an independent measurement

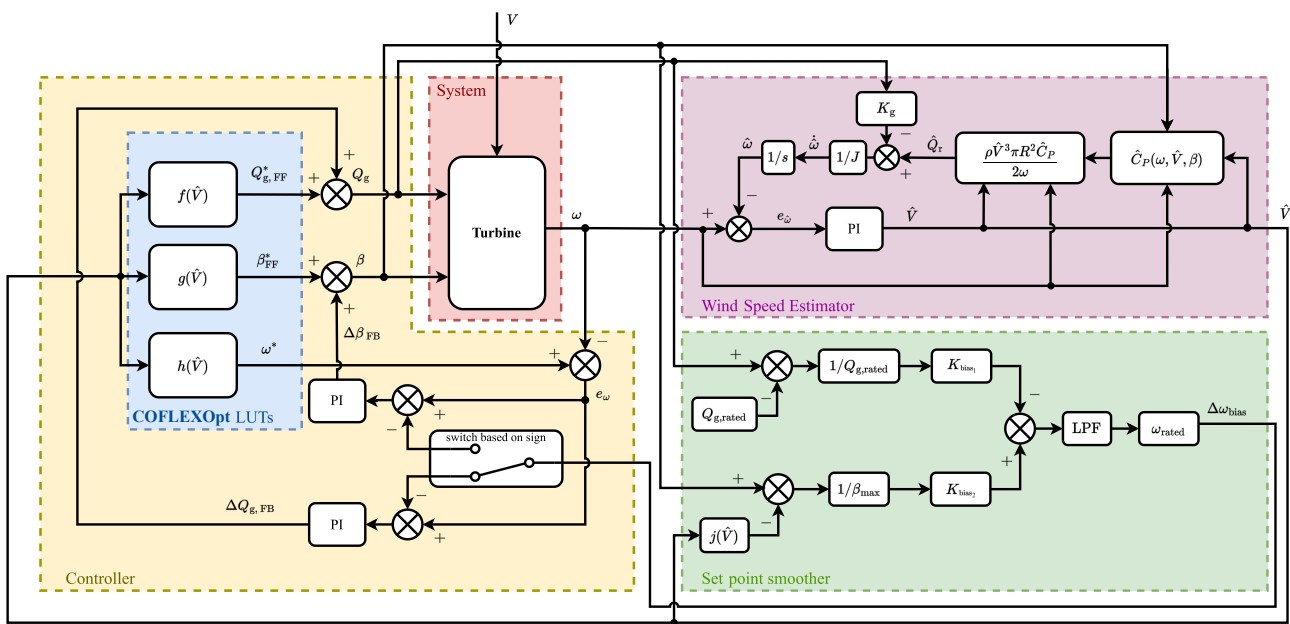

**Figure 14.** Block diagram of the novel control scheme for large, flexible wind turbines, showing the integration of the wind speed estimator, set point smoothing technique, and feedforward-feedback controllers.

of the rotor-average wind speed—such as LiDAR measurements—or by combining such measurements with the estimated
values. Alternatively, one can update the aerodynamic model (used in both the controller and estimator) to represent the actual, possibly degraded, aerodynamic properties of the wind turbine using online learning algorithms (Mulders et al., 2023).

    The capabilities of this novel scheme to allow for a smooth transition between the two PI controllers are visualised in Fig. 15, where the behaviour of the controller is analysed in a time-domain simulation. This analysis was performed using the characteristics of the flexible model described in Sect. 2, in the time-domain wind turbine aeroelastic simulator HAWC2. In
this example, the controller tracks the optimised set point strategy defined by *Case 2*. The transition between the partial-load and full-load controllers is expected to occur at a wind speed of approximately 10.5 metres per second (see Fig. 7). To observe this transition in detail, we have extracted a 40-second segment from a 1000-second simulation carried out with a turbulent wind field and wind shear, capturing the moment when the rotor's average wind speed crosses the rated wind speed.

    Figure 15 (a) compares the rotor-average wind speed (light grey) with its corresponding estimate (dark grey). Overall, the
two signals align well, though the estimated value shows some high-frequency oscillations that likely stem from noise in the WSE input signals and the calibration of the WSE. Brief discrepancies also occur (e.g. near $t \approx 510$ s), which may be attributed to dynamic effects or degrees of freedom not captured by the internal model used in the WSE. To prevent the high-frequency oscillations from directly exciting the actuators, we apply a first-order low-pass filter with a cut-off frequency of $0.5\pi$ rad s$^{-1}$ to the feedforward inputs. Figures 15 (b) and (c) show, respectively, the feedforward pitch and torque commands scheduled on the
true rotor-average wind speed (light grey), on the estimated wind speed (dark grey), and the actual controller outputs (green).

Up to $t \approx 505$ s, the turbine remains in partial-load operation: The collective pitch angle closely follows the feedforward command, which in turn tracks the ideal feedforward value reasonably well. Near $t = 505$ s, the generator torque saturates (Fig. 15 (c)) to maintain rated power. At that moment, the estimated wind speed in Fig. 15 (a) reaches around $10.7$ m s$^{-1}$, matching the expected rated condition. Figure 15 (d) illustrates how the set-point bias $\Delta \omega_{\mathrm{bias}}$ (blue) ensures a smooth transition from torque to pitch control. Before $t \approx 505$ s, the bias is negative, keeping the collective pitch angle saturated at its lower limit and allowing the torque controller to be active. As the system approaches rated, the bias crosses zero and effectively drives the generator torque into saturation, activating the collective-pitch controller. This gradual shift avoids abrupt changes in control action and demonstrates that the combined feedforward-feedback strategy can successfully handle transitions to full-load operation, even under turbulent inflow. Finally, while the overall dynamic performance is satisfactory, further gain scheduling or fine-tuning of the WSE and PI loops could improve transient behaviour and reduce any remaining high-frequency pitch or torque activity.

The next section will delve deeper into the validation of the proposed COFLEX control scheme through time-domain simulations with uniform wind step inputs and turbulent wind fields.

## 6 Results

In this section, we present the results of time-domain simulations carried out to verify the effectiveness and robustness of the newly developed control strategy for large, flexible wind turbines. In Sect. 6.1, we assess the COFLEX control scheme using uniform wind steps simulations. Step responses are commonly used in controller design to evaluate dynamic transient response, particularly in terms of performance and stability. In our case, these tests serve multiple purposes: to verify the controller functionality of the controller across the full operating range, including partial and full load regions as well as the transition between them; and, most importantly, to confirm that the operational strategy defined by COFLEXOpt mappings is consistently maintained through the proposed control scheme, as evaluated with full aeroelastic HAWC2 simulations. Then, in Sect. 6.2, we analyse the simulations carried out with turbulent wind fields to test the controller under more realistic operating conditions for a wind turbine.

The following subsections will detail the specific simulation setups, the methods used to analyse the performance, and the comparisons made with reference values obtained from COFLEXOpt results shown in Fig. 7. The model used for the time-domain simulations is equivalent to the **Flexible** model described in Table 1. The tool used to perform simulation is HAWC2, a mid-fidelity aeroelastic code capable of handling coupled structural deformations of the blades, which was already employed to calculate the performance of the same wind turbine in Rinker et al. (2020).

### 6.1 Time-domain simulations: uniform wind cases

Four 2500-second simulations were carried out with incremental wind speed steps of one metre per second every 100 seconds, starting from an initial wind speed of 3 metres per second up to 25 metres per second, for each set points strategy. These

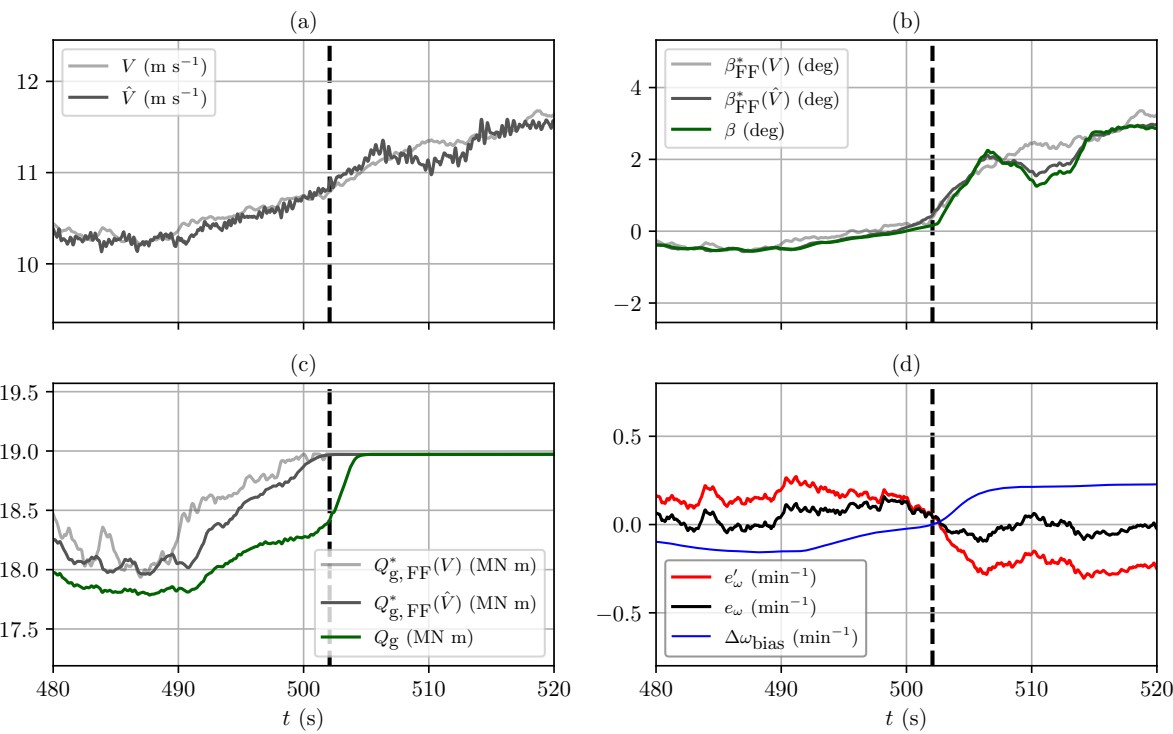

**Figure 15.** Quantities extracted from a time domain simulation of the IEA 15 MW RWT with turbulent wind and wind shear, performed in HAWC2 with the implementation of the novel control scheme, showing the behaviour of control inputs and set-point smoothing technique values near the transition from partial load to full load. The vertical dashed line at $t \approx 505$ s marks the transition from generator torque control to collective pitch control in the full-load region. (a) Rotor-average wind speed (light grey) and estimated wind speed (dark grey). (b) Ideal feedforward collective pitch angle scheduled on the actual rotor average wind speed (light grey), feedforward scheduled on the estimated wind speed (dark grey), and the controller pitch command (green). (c) Ideal feedforward generator torque scheduled on the actual rotor average wind speed (light grey), feedforward input scheduled on the estimated wind speed (dark grey), and the actual generator torque command (green). (d) Rotational-speed error $e_\omega$ (black), biased error $e'_\omega$ (red), and the set-point smoothing technique bias $\Delta\omega_{\text{bias}}$ (blue).

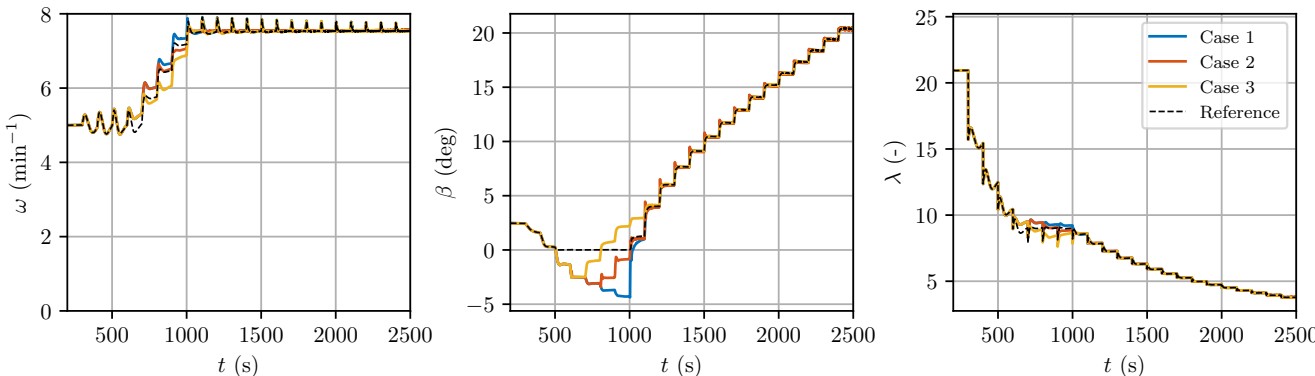

**Figure 16.** Time series of rotational speed, collective pitch angle, and tip-speed ratio for the four different strategies, in time-domain simulations performed with HAWC2, with uniform wind steps. The plots demonstrate the control system's ability to follow different set point strategies and maintain stable operation across varying wind speeds.

simulations, which included an initialisation period of 200 seconds to settle down transient behaviour, were used to test the controller step response.

The time series of rotational speed, collective pitch angle and tip-speed ratio are shown in Fig. 16, excluding the initialisation
period. In the first 500 seconds, all control strategies correctly track the minimum rotational speed ($5 \ \mathrm{min}^{-1}$), with relatively high overshoots (around 10%), while the collective pitch angle is set to the same values to maximise the power coefficient. From $t = 500$ s onwards, the four cases follow different set point strategies for both rotational speed and collective pitch angle. In all cases, varying trends on the overshoot and settling time values suggest that gain scheduling (see Abbas et al., 2022) could be employed to improve the dynamics of the controller and is devoted to future work. The focus of this paper is to establish a
novel control strategy for flexible turbines, and therefore, we are interested in analysing the trends in steady-state performance.

During the simulation, the final 10 seconds for each 100-second interval were used to calculate the steady states of selected variables. These steady states (dots), obtained using the FF-FB control scheme in HAWC2 simulations, are presented in Fig. 17 w.r.t. the input wind speeds (uniform and constant) in the same intervals and compared to the prescribed operating points (lines) calculated through COFLEXOpt.

These results demonstrate that, in time-domain simulations, COFLEX accurately tracks the set points calculated with COFLEXOpt for all variables of interest. The trends observed in the mean values of control variables, including rotational speed, collective pitch angle, and generator torque, follow the strategies prescribed by COFLEXOpt. This novel approach with a variable tip-speed ratio and collective pitch angle allows for maximising power production while respecting the blade tip displacement constraint.

The generator torque reaches saturation above $11 \ \mathrm{m\,s}^{-1}$ for all cases, except *Case 3*, in which the tighter constraint on tip displacement increases the wind speed at which the rated power is produced up to $12 \ \mathrm{m\,s}^{-1}$. The steady-state maximum values of tip displacement for *Case 2* and *Case 3* result in 14.1 metres (+3.5 %) and 10.6 metres (+6.0 %), respectively, showing

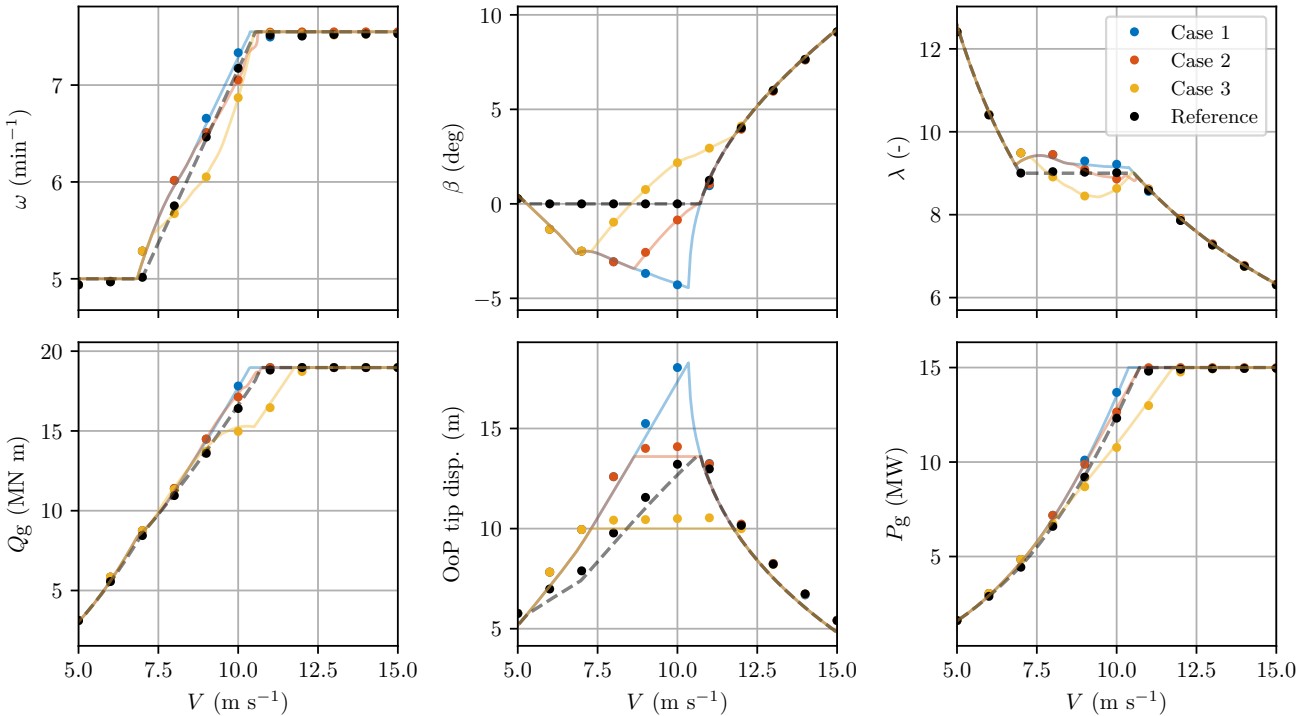

**Figure 17.** Comparison of steady states (**dots**) calculated from the time-domain HAWC2 simulation and prescribed operating points (**lines**) from COFLEXOpt based on HAWCStab2 linearisations for the four different strategies. Steady-state trends match the expected operating points, meaning that the novel controller is able to track the set points for the entire operating range of the IEA 15 MW RWT.

a positive slight discrepancy. For all cases and each wind speed, the OoP tip displacement is slightly underestimated in the steady states calculated through COFLEXOpt. This small difference can be attributed to different factors: a discrepancy in the

steady-state blade deflection calculation for HAWC2 and HAWCStab2, which was deemed small but not directly quantified in the comparison of the tools (Verelst et al., 2024); nonlinear dynamic effects which are only taken into account by HAWC2.

Some differences are also present in the generator torque and collective pitch angle set points (see, e.g. *Case 3* generator torque and *Case 1* collective pitch angle at wind speeds near rated). These differences can be attributed to the activation of the switching logic and the resulting set-point bias affecting the system behaviour. The potential of the new control scheme to

track them in realistic turbulent wind conditions is provided in the next section with the analysis of turbulent wind cases.

### 6.2   Time-domain simulations: turbulent wind cases

To evaluate the performance of the controller under more realistic operating conditions, turbulent wind cases were defined following the design load case (DLC) 1.1 as specified in IEC 61400-1 for wind class IB (International Electrotechnical Commission, 2019). A series of 1000-second simulations were performed with mean wind speeds ranging from 3 to 25 metres per

second (one simulation every metre per second) and turbulence intensity in accordance with IEC standards, using six different seeds for the turbulence box generator (for a total of 138 simulations for each control strategy). The turbulent wind fields were generated using the Mann turbulence box generator integrated within HAWC2. Additionally, a power-law vertical wind shear was applied with an exponent of 0.2.

For each simulation, only the last 600 seconds were used in the analysis to eliminate initialisation dynamics. Simulations were grouped for each control strategy and subdivided into small time intervals of 10 seconds. Then, the means of the individual performance metrics were calculated in these intervals and binned with respect to the average wind speed of the rotor, with a uniform bin length of one metre per second.

A statistical analysis was performed, and the distribution of selected performance metrics is shown for the control strategies *Case 1* and *Case 2* in Fig. 18 and Fig. 19, within a wind speed range of 5 to 15 metres per second. In these figures, the dotted lines represent the prescribed operating points from COFLEXOpt for the same control strategy, calculated using the rotor average wind speed $V$, while the dashed grey lines (corresponding to the right y-axis) indicate the differences between the median values for each bin and the prescribed values at each bin mid-point. The boxplots depict the distribution of the performance metrics averaged over ten-second intervals for each wind speed bin, with indications of quartiles (the filled boxes with a central line represent the 25%, median, and 75% quartiles), minimum and maximum values (whiskers limits), and outliers (triangles).

For both strategies, the median values of the estimated wind speed ($\hat{V}$) are consistently higher than the actual wind speed. This discrepancy is around 10% at very low wind speeds, decreases to approximately 5% near the rated wind region, and then increases again linearly in the full load region. This consistent, positive bias was not observed in previous analyses and is likely driven by local wind speed fluctuations due to wind shear and turbulence. Our wind speed estimator uses a torque-balance approach, matching the measured generator torque to an estimated rotor torque, recalling the system of Eq. 10. Under wind shear and turbulence, the contribution of blade sections to the total torque depends on the local velocities. Hence, the effective wind speed which produces the rotor torque differs from the arithmetic mean across the rotor disk. As a result, the WSE estimates an effective wind speed that differs from the rotor average wind speed, which is used as a reference here. However, this bias does not degrade the performance of the controller. In a practical scenario, the controller must adapt to this effective wind speed; the control scheme of COFLEX still holds, as our set-point mappings and feedforward inputs rely on precisely this torque-based wind speed estimate.

Both the collective pitch angle and generator torque exhibit differences relative to the set points, with similar magnitudes and trends across the two analysed control strategies. These differences can be largely attributed to the bias between the estimated wind speed and the rotor average wind speed resulting from the simulator used for binning. This directly impacts the feedforward component in the control loop, especially at low wind speeds. Despite these discrepancies, the median values of the OoP tip displacement closely follow the steady-state values calculated by the set point optimiser, with a high degree of accuracy (less than 10% difference across the analysed operating range). In *Case 2*, the expected constraint on the median value of the OoP tip displacement is satisfied with a deviation of less than 1%.

While constraining the steady-state OoP tip displacement helps reduce average deflection levels, more advanced control techniques remain necessary to mitigate the transient effects that drive the maximum values—and thus the tower-strike risk. Consequently, imposing a strict limit on the maximum displacement would require a different control approach, such as online set-point optimisation (Petrović and Bottasso, 2017) or advanced individual pitch control (Liu et al., 2022a), which can explicitly predict and counteract such extremes. Nonetheless, to address the safety margin in a stochastic way, one could modify the constraint in COFLEXOpt by incorporating a precomputed variance around the median displacement. This would allow designers to ensure, a priori, that the probability of exceeding the maximum allowable OoP tip displacement remains within an acceptable margin.

The generator torque is correctly saturated in the full load region for both strategies. Finally, we observe an interesting effect on the generator power median values in the partial load region, where these values consistently exceed the prescribed operating points. These trends align with studies on the effects of turbulence intensity on the power production of wind turbines in the partial load region (Saint-Drenan et al., 2020).

Figure 20 compares the median values of OoP tip displacement (top panel) and generator power (bottom panel), both normalised by the reference strategy, for the new strategies across wind speeds from 5 to 15 metres per second. For *Case 1* and *Case 2*, we observe that the generator power increases by approximately five percentage points relative to the reference at the expense of higher tip displacements in the partial load region. In particular, *Case 1* shows OoP tip displacements as much as 30% above the reference at rated wind speed, which aligns with the prescribed operating points. In *Case 2*, the displacement constraint is active around $10 \, \mathrm{m\,s^{-1}}$, as indicated by the orange bars converging toward unity in the top panel near $11 \, \mathrm{m\,s^{-1}}$. *Case 3* follows a similar pattern at lower wind speeds (below $8 \, \mathrm{m\,s^{-1}}$), but the tighter constraint on tip displacement results in values around 25% below the reference near the rated wind speed, and a corresponding lower power output in that range. All three cases behave similarly to the reference controller in full-load operations. Overall, these trends confirm that the set points derived via COFLEXOpt can be effectively tracked in turbulent inflow scenarios.

## 7   Conclusions

This work introduces COFLEX: a novel set point optimisation and control strategy for large, flexible wind turbines, addressing the limitations of conventional methods. Unlike traditional strategies that rely on fixed tip-speed ratio and fixed collective pitch angle, our approach optimises set points for varying rotational speed, pitch angle, and generator torque across the turbine's full operational range without the need to predefine operating regions.

The first module of COFLEX, the set point optimiser COFLEXOpt, was used to obtain new control strategies for the IEA 15 MW RWT turbine and compare them to the reference fixed tip-speed ratio tracking scheme. Using COFLEXOpt, we derived variable tip-speed ratio and collective pitch angle schedules for power maximisation, with and without constraints on blade out-of-plane tip displacement. In one of the analysed cases, we achieved up to an 8% increase in generator power across the partial-load region compared to the reference strategy. Additionally, we demonstrated the ability to incorporate constraints on structural and operational requirements in COFLEXOpt, such as limiting out-of-plane tip displacement. For the case where

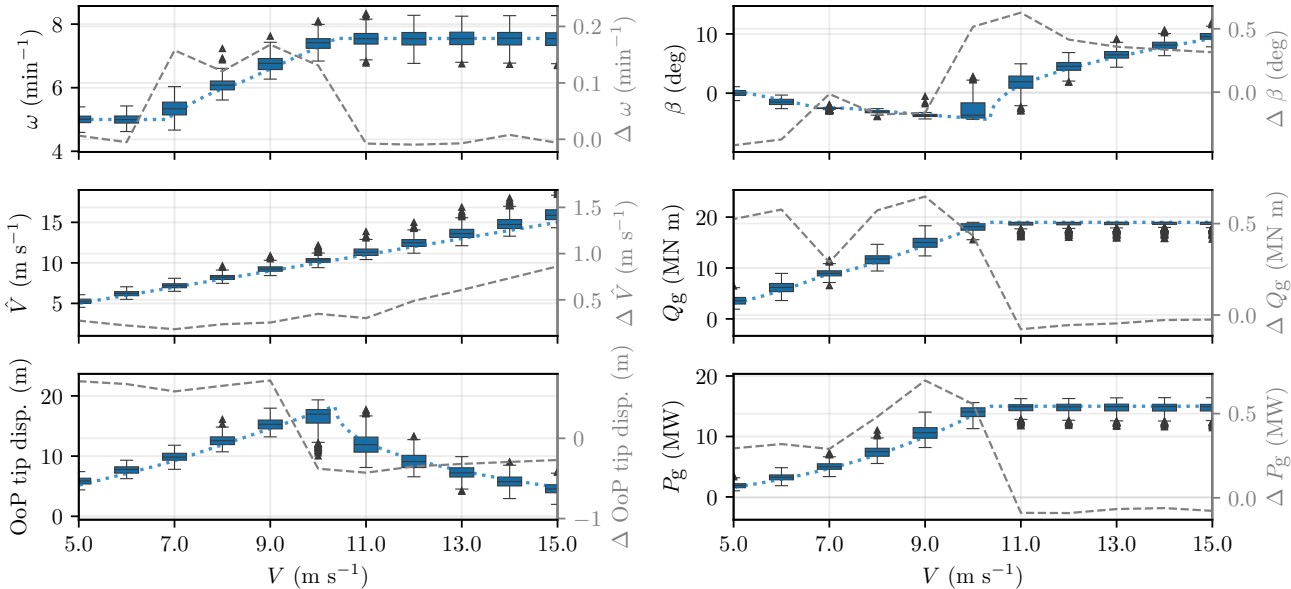

**Figure 18.** Statistical analysis of performance metrics under turbulent wind conditions for the *Case 1* control strategy. The boxplots represent the distribution of average performance metrics over 10-second intervals, categorised into wind speed bins of one meter per second. The filled boxes indicate the 25th, 50th (median), and 75th percentiles, with whiskers extending to the minimum and maximum values of the distribution and triangles marking outliers. The dotted lines correspond to the prescribed operating points from COFLEXOpt for the same control strategy, while the grey dashed lines (associated with the right y-axes) represent the errors between the median values and the optimiser's set points for each wind speed bin midpoint. Although a slight discrepancy in wind speed estimates is present, the deviations between median values and prescribed operating points remain minimal across other metrics.

the blade deflection limit matched the maximum value from the reference strategy, we still observed an increase in generator power of about 5%.

A feedforward-feedback controller was designed to track the optimised set points, relying on a new, more accurate wind speed estimator algorithm that uses three-dimensional $C_P$ tables, parameterised on rotational speed, wind speed and collective pitch angle. A set point smoothing technique was developed to allow for a seamless transition between partial and full load operations of a wind turbine.

Time-domain simulations were employed to validate the capabilities of the controller under various wind conditions and in the transition region. The wind step response simulations indicated that the controller effectively reached the steady states prescribed by COFLEXOpt schedules across the entire operating range and that it was able to operate smoothly in the transition region. A statistical analysis of the performance of the control scheme under turbulent wind conditions was carried out to evaluate its robustness in more realistic operating scenarios. Selected performance metrics were analysed in the operating range with a mean wind speed from 5 to 15 metres per second. Despite a slight wind speed estimation bias, which may be

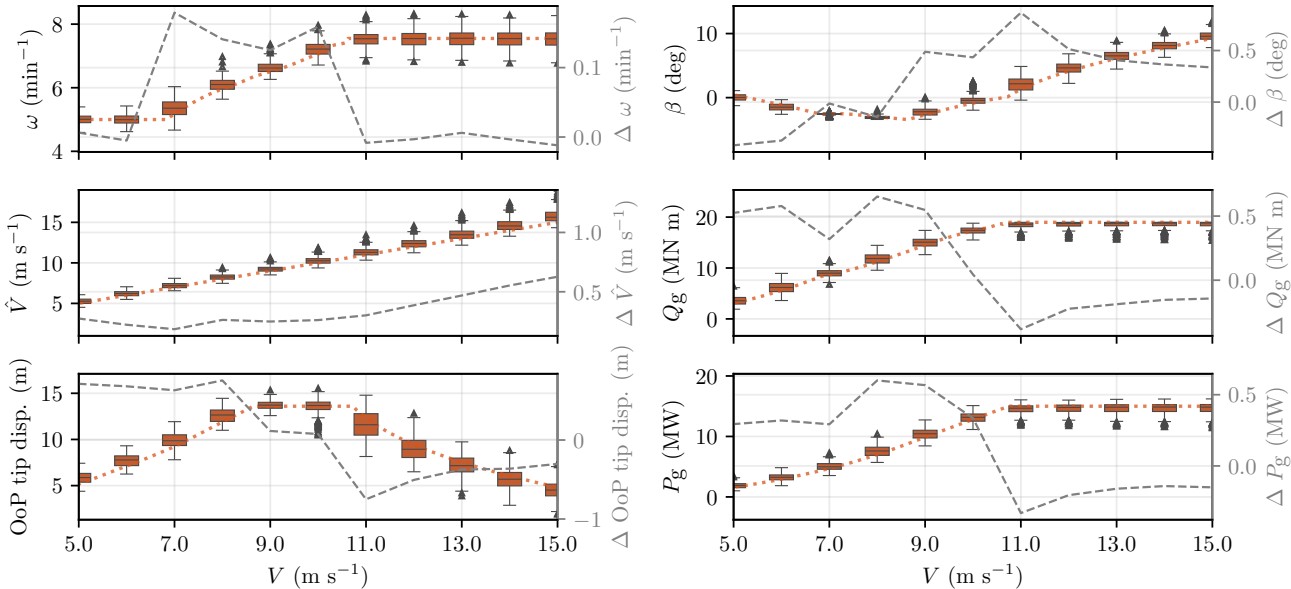

**Figure 19.** Statistical analysis of performance metrics under turbulent wind conditions for the *Case 2* control strategy. The boxplots represent the distribution of average performance metrics over 10-second intervals, categorised into wind speed bins of one meter per second. The filled boxes indicate the 25th, 50th (median), and 75th percentiles, with whiskers extending to the minimum and maximum values of the distribution and triangles marking outliers. The dotted lines correspond to the prescribed operating points from COFLEXOpt for the same control strategy, while the grey dashed lines (associated with the right y-axes) represent the errors between the median values and the optimiser's set points for each wind speed bin midpoint. As demonstrated by the small deviations between median values and prescribed operating points in all control and output variables, COFLEX confirms its suitability for real-world scenarios, where compliance with constraints is essential.

695 attributed to the difference in the estimated effective wind speed and rotor average wind speed, the controller maintained tracking of rotor speed, generator torque, and collective pitch angle under turbulent conditions. The median values of rotor speed across different wind speeds were generally contained within a small margin (with an error of 5%) with the desired set points. Generator torque and collective pitch angle outputs were similarly accurate, with small deviations.

Moreover, the controller effectively achieved the expected out-of-plane tip displacement and generator power steady states across different wind conditions. The analysis showed that the out-of-plane tip displacement and generator power closely

700 tracked the optimised set points derived from COFLEXOpt. The ability to reach the desired steady states highlights the potential of the novel control scheme to enhance performance while complying with structural integrity in large, flexible wind turbines.

Looking forward, this framework could be leveraged for the co-design of large, flexible wind turbines, integrating structural and control variables from the earliest design stages. Additionally, the control scheme of COFLEX can be adapted to perform online set-point optimisation to limit the maximum values reached in dynamic situations—such as wind gusts—that

705 can suddenly increase out-of-plane tip displacement.

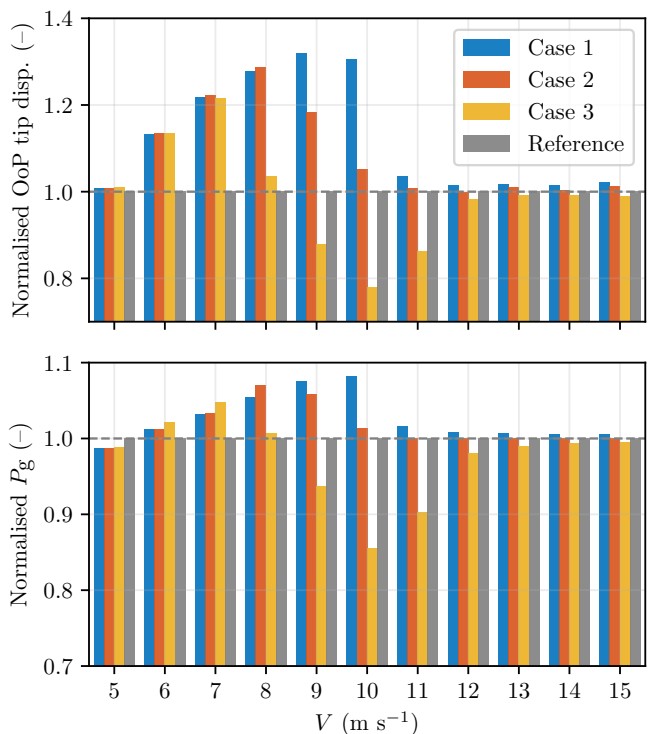

**Figure 20.** Median out-of-plane tip displacement (*top*) and generator power (*bottom*), both normalised by the values obtained with the reference strategy for each wind speed bin across wind speeds of 5 to 15 m s$^{-1}$. Bars represent 10-second median values obtained from six 600-second HAWC2 simulations under realistic turbulence, grouped in 1 m s$^{-1}$ bins. The reference strategy values (unity) are shown in grey, while *Case 1* (blue), *Case 2* (orange), and *Case 3* (yellow) bars represent the values obtained with the new strategies. In *Cases 1* and *2*, power increases relative to the reference, but tip displacements rise by up to 30% in partial-load operation. *Case 3* exhibits a 25% reduction in tip displacement near rated wind speed, associated with generally lower generator power.

*Code and data availability.* Code and data are available at the public repository (Lazzerini et al., 2024)

*Author contributions.* GL, JDG, TGB, FC and SPM worked at conceptualizing this research and establishing the methodology. GL, JDG, TGB, and SPM developed COFLEX and COFLEXOpt. GL ran all simulations. GL and VDM post-processed data from the simulations. DDT and SPM provided feedback on the methodology. GL prepared the original draft with contributions from all authors, which was then reviewed by all authors.

*Competing interests.* The authors declare that they have no conflict of interest.

710

*Acknowledgements.* The authors acknowledge the contribution to the development of COFLEX by Markel Meseguer San Martin and Ebbe Nielsen from Shanghai Electric Wind Power Generation European Innovation Center. This research received funding from TKI Wind op Zee, under grant "TKITOE WOZ 2309 TUDELFT COCOFLEX".

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
