# Peer review of "COFLEX: A novel set point optimiser and feedforward-feedback control scheme for large flexible wind turbines"

_Wind Energy Science, 2024_

## Author Comment (AC2)

| | |
|---|---|
| Date | March 21, 2025 |
| Our reference | n/a |
| Your reference | wes-2024-151 |
| Contact person | Guido Lazzerini |
| E-mail | g.lazzerini@TUDelft.nl |
| Subject | Author's Response |

**Delft University of Technology**

Delft Center for Systems and Control

Address
Mekelweg 2 (ME building)
2628 CD Delft
The Netherlands

Reviewers
*Wind Energy Science Journal*

Dear reviewers,

The authors would like to thank the reviewers for the constructive and thorough comments and suggestions for our paper. We believe that your feedback has helped us significantly improve the quality of the manuscript.

To consider all the feedback, the paper has been carefully revised. The objective of this document is to reply to the points raised and provide a detailed overview of the changes made. For each comment, a point-to-point response is provided in blue color, while the corresponding changes to the manuscript are reported in red. Please note that, in the enclosed marked-up version of the revised manuscript, the removed and added portions of the manuscript are indicated by red strikethrough text and blue underlined text, respectively. We hope that this document provides satisfying answers to the reviewers' comments.

Yours sincerely,

Guido Lazzerini
Jacob Deleuran Grunnet
Tobias Gybel Hovgaard
Fabio Caponetti
Vasu Datta Madireddi
Delphine De Tavernier
Sebastiaan Paul Mulders

Enclosure(s):    Response to Reviewer 1
Response to Reviewer 2
Marked-up version of the revised manuscript

**Response to Reviewer 1**

**General Comments**

The manuscript presents a novel control strategy, COFLEX, for large flexible turbines, addressing a critical gap in optimizing turbine performance considering structural flexibility. This study is highly relevant as the increasing scale of wind turbines introduces significant challenges in structural dynamics and control. The integration of a set point optimization framework (COFLEXOpt) with a feedforward-feedback control scheme represents a substantial improvement over traditional tip-speed ratio-based methods.

The manuscript effectively builds upon existing methodologies and tools for optimization, estimation, and control. While none of the individual modules (optimization framework, wind speed estimator, or controller) are entirely new, the parameterization of dimensionless coefficients in three dimensions instead of two is a noteworthy adaptation. These adaptations are well-discussed, seamlessly integrated, and tested comprehensively through simulations. The paper is clear, well-structured, and includes detailed methodologies and results.

**Response:** Thank you for your kind words and appreciation of our work. We also thank you for your feedback which helped improving the work further. In the following section, we provide our responses to your specific comments.

**Specific Comments**

1. Design of Wind Turbines:

   The manuscript could benefit from a brief discussion of passive design techniques such as pre-coning/pre-bending of blades to mitigate flexibility effects and tailored bend-twist coupling for passive load alleviation.

   **Response:**
   Thank you for your suggestion. A brief discussion of passive design techniques indeed improves situating our work in a broader perspective of passive and active techniques to alleviate loads while optimizing performance. Therefore, we added a paragraph to the introduction.

   **Revised portion:**
   - **Line 37 - 42:**
     "...Passive design techniques such as pre-coning, pre-bending, and bend-twist coupling can mitigate some of these effects by modifying the geometrical

and structural properties of the rotor. For instance, while pre-coning and pre-bending can increase blade-to-tower clearance and increase the maximum swept area when the turbine is operating at its rated condition, bend-twist coupling can be used to reduce aerodynamic loading passively (Sartori et al., 2018). Nonetheless, these structural measures remain complementary to advanced active control, which can further optimise energy capture and help decrease loads (Bortolotti et al. 2019). ... "

2. Objective Function:

The motivation for solely minimizing the torque and power coefficient in the steady-state optimization is unclear.

It is recommended to
(a) describe the optimization framework in a more general form
(b) include a discussion on other meaningful objectives to demonstrate versatility of the proposed framework.

The use of a single objective function across the entire operating region is intriguing as objectives are considered to vary between full and partial load operation.

**Response:**
Thank you for your valuable feedback on this point. In the original manuscript, we included a general form of the optimisation problem required to compute the set points shown in Eq.(6). This general formulation shows that the objective $f_{\mathrm{obj}}$ and the constraints $\mathbf{C}_{\mathrm{eq}}, \mathbf{C}_{\mathrm{iq}}$ may be chosen freely by the user/designer, enabling the framework to handle alternative objectives or terms. We acknowledge the lack of sufficient description of this general representation, so we added it right after the general form of the problem.
In the revised manuscript we clarify the possible choices for the objective function and constraints. The choice of maximising the power coefficient in the objective function is obvious. Additionally, we introduced the rationale behind minimising the torque coefficient. Essentially, the solution to the NLP is not unique in the full-load region (i.e. $\overline{V} > V_{\mathrm{rated}}$) because there are infinitely many pairs $(\omega^*, \beta^*)$ that all return the power coefficient to produce the rated power. This was shown graphically in the lower-left panel of Fig. 6 of the original manuscript, where solutions valid for the rated power can be found on the red iso-line. To find a unique solution to the general problem that is valid for the entire operating range, the objective function needs to "select" a single (optimal) point on that red iso-line. Mathematically, this means including an additional term that produces different iso-lines in the full-load region so that the objective function is convex and returns one unique solution. From a design perspective, that extra term must "compete" with the power coefficient. If one uses it merely for regularisation, a straightforward choice is a term that remains sufficiently small in the partial-load region—thus avoiding sub-optimal power capture—while being large enough to enable finding a unique solution in full load. Competing objectives for the power coefficient include aerodynamic torque or thrust, among others.

As a result, we include only a small regularisation term (i.e., the torque coefficient), ensuring limited impact on partial-load solutions. Furthermore, we prefer to treat structural deformations as non-linear constraints. By doing so, we can directly control the maximum (steady-state) blade deflections (and, indirectly, loads).

A comparable methodology to find optimal set-point schedules was described by Iori et al. (2022), which we have now cited in the revised manuscript. To this point, we also added a brief discussion of potential additional objectives that can be integrated into COFLEXOpt, which shows its versatility.

**Revised portion:**

- **Lines 300 - 321:**
  "... The versatility of this framework lies in the wide range of possible definitions for $f_{\mathrm{obj}}$, $\mathbf{C}_{\mathrm{eq}}$, and $\mathbf{C}_{\mathrm{iq}}$. In particular, $\mathbf{C}_{\mathrm{eq}}$, and $\mathbf{C}_{\mathrm{iq}}$ can include any metrics representable in the $(\omega, V, \beta)$ space. Since the tip-speed ratio is decomposed into two separate variables, one can incorporate non-linear constraints dependent on actual operating conditions. Examples include structural deflections, peak thrust (as in peak-shaving strategies), load-alleviation targets (e.g. bounding the root flapwise bending moment), or blade-span-dependent quantities (e.g. limiting angle of attack or relative velocities). Regarding the objective function $f_{\mathrm{obj}}$, its formulation must yield unique and optimal solutions $(\omega^*, \beta^*)$ across the entire operating range. The primary objective is to maximise power capture (i.e. the power coefficient). The power output will also naturally be subject to an inequality constraint, ensuring the rated power is not exceeded. However, once the rated power limit is reached in the full-load region (i.e. $\overline{V} > V_{\mathrm{rated}}$), infinitely many $(\omega^*, \overline{V}, \beta^*)$ combinations yield the power coefficient to produce the rated power and the maximisation of the power coefficient is not sufficient to produce unique solutions.

  To address this, we introduce a secondary term in the objective function, resolving the non-uniqueness of the solution. This technique, also suggested in Iori et al. (2022), selects one point along the power coefficient iso-lines based on the minimisation of a secondary term in the objective function, resolving the non-uniqueness of the solution. In particular, this secondary term can have physical meaning: for example, if one selects the thrust coefficient, an increase in rotor loading is penalised in the optimal solution. Alternatively, one can penalise the torque coefficient, which ensures that the optimizer seeks the solution that yields the lowest rotor torque within the feasible region—helping to mitigate drivetrain loading. If the weight on this secondary term is kept sufficiently small, it effectively acts as a regularisation term while still retaining power maximisation as the primary objective. In our case, having defined custom inequality constraints that can include loads and structural deformations, we can directly target load alleviation through the imposition of limit (steady-state) values. As a result, we include only a small regularisation term in the objective function to ensure a limited impact on partial-load solutions. Hence, we propose maximising the power coefficient with a penalisation on the rotor torque coefficient for each wind speed $\overline{V}$, as follows: ..."

3. Model Limitations:

A discussion on robustness of the proposed control approach is recommended, including potential discrepancies between HAWC2 simulations and real-world turbine dynamics.

How do modeling uncertainties and measurement errors affect performance?

**Response:**
Thank you for raising this point. In the original manuscript, we cited Brandetti et al. (2022) to illustrate how inaccuracies in the power coefficient tables affect wind speed estimates (see Section 5.1). In the revised manuscript, we have added a brief discussion in Section 5.4 that analyses the overall robustness of the combined control scheme.
Specifically, modelling uncertainties in the internal aerodynamic model can lead to biased wind speed estimates, which in turn result in biased feedforward inputs. Although the feedback components work to drive the system back to a reference rotor speed, denoted as $\omega^*$, this reference is also affected by the biased wind speed estimate $\hat{V}$. Consequently, in case of modelling uncertainties, the tracked operating point does not exactly reflect the intended optimal behaviour of the turbine. Similarly, measurement errors can propagate in the wind speed estimate, leading to biased tracking. A potential solution to this issue would be to base the feedforward input on an independent measurement of the rotor-average wind speed, for instance, by using lidar measurements or a combination of lidar measurements and estimated values, thereby mitigating the effects of the bias. Another approach is to update the aerodynamic model (used in the controller and estimator) based on the wind turbine's current aerodynamic properties. Our research group has proposed two learning methods for this: one excitation-based and another excitation-free, relying on wind speed measurements.

**Revised portion:**

- **Lines 538 - 552:**
  " ...This control scheme leverages feedforward action to achieve the desired set points, while feedback loops work to enhance stability, correct (tracking)

errors, and add resiliency to disturbances and noise. However, its overall tracking performance is dependent on the accuracy of the internal power coefficient table. The wind speed estimation relies on this table, so any bias in the power coefficient data propagates into the estimates. As demonstrated by Brandetti et al. (2022), for the WSE-TSR tracking scheme, whenever the controller's reference is scheduled based on wind speed estimates, the system converges to a steady state that reflects this bias. In other words, the controller is capable of tracking a reference, but the reference itself is shifted from the true optimal operating point. This is essentially the same phenomenon encountered in standard tip-speed ratio tracking, where the *optimal* set point is also calculated offline using nominal aerodynamic data; if the real performance deviates from that nominal data, the turbine will no longer be operating at the true optimum. Our scheme will similarly be affected by inaccuracies in the internal power coefficient table, even though it maintains effective reference tracking. A potential mitigation of the bias introduced by modelling inaccuracies would be to schedule the feedforward input on an independent measurement of the rotor-average wind speed—such as lidar—or by combining such measurements with the estimated values. Alternatively, one can update the aerodynamic model (used in both the controller and estimator) to represent the actual, possibly degraded, aerodynamic properties of the wind turbine using online learning algorithms (Mulders et al., 2023). ..."

4. Wind Speed Estimator:

The wind speed estimator described in Section 5.1 is tested using a $K\omega^2$ control law.

Please clarify which power coefficient is used in the $K\omega^2$ law and discuss if/how the selection of the feedback gain $K$ impacts the estimator's performance.

**Response:**
Thank you for highlighting this point. We wish to clarify that the $K\omega^2$ controller used in this section serves only as a convenient means to evaluate the wind speed estimator (WSE) performance in an "open-loop" fashion, where the resulting rotor speed, pitch, and torque control signals are directly fed to the WSE while the output wind speed estimate does not affect the control routines. For all other sections of the paper, our advanced COFLEX controller is used in closed-loop with the newly adapted WSE. The constant K for the $K\omega^2$ scheme was calculated based on the optimal tip-speed ratio prescribed by the IEA 15 MW baseline design. As a result, the system reaches steady states that do not depend on the WSE and the estimator performance is then evaluated at that point. Although the feedback gain $K$ affects the dynamic behavior of the estimator, our analysis of the steady-state bias in wind speed estimation confirms is unaffected by this value. We have revised

the corresponding paragraph in Section 5.1 accordingly.

**Revised portion:**

- **Lines 437 - 445:**
  "...To verify the improved performance of the WSE with an additional power coefficient table dimension, three time-domain simulations of the IEA 15 MW RWT were performed with uniform wind steps of $1\,\mathrm{m\,s^{-1}}$ ranging from 3 to 11 $\mathrm{m\,s^{-1}}$, each step lasting 300 s. To analyse the accuracy of the steady-state wind speed estimation, we implemented a $K\omega^2$ scheme, selecting the gain $K$ according to the method in Pao and Johnson (2011). The constant $K$ was calculated based on the optimal tip-speed ratio prescribed by the IEA 15 MW RWT baseline design, reverting to the standard constant optimal tip-speed ratio assumption. The constant $K$ was calculated based on the optimal tip-speed ratio and corresponding maximum power coefficient prescribed by the IEA 15 MW RWT baseline design, reverting to the standard constant optimal tip-speed ratio assumption. In doing so, the steady-state behaviour is fully specified by the gain $K$ so that the generator torque controller does not rely on wind speed estimates. This approach decouples the steady-state performance of the WSE from other control routines, allowing us to evaluate the estimator without interference from the control tuning parameters. The $K\omega^2$ controller used in this section serves only as a convenient means to assess the WSE steady-state performance. ..."

5. Saturation Schedules:

A clearer motivation is needed why saturation limits are computed solving the "reduced" optimization described in Equation (18).

Could alternative formulations also be used?

**Response:**
Thank you for this feedback. First, let us clarify that we needed to find a general way of obtaining the lower pitch saturation schedule that could be integrated with COFLEX (and COFLEXOpt optimized set points). First, we noticed that in Abbas et al. (2022), the so-called "minimum pitch schedule" of Fig. 9 is said to produce set points for "peak shaving and power maximization in low wind speeds." However, in their work, it is not fully clear how this schedule was obtained.
For our needs, it seemed natural to adapt the set point optimiser to find these schedules; We made the following considerations to build the reduced optimization problem of Equation (18):

- The lower pitch angle saturation value needs to produce an *aerodynamically*

*stable* point for the operations of a wind turbine in full load. That is the lowest collective pitch angle that can be reached for a given wind turbine wind speed and given rotor speed should let the blades stay away from stalling. This point led to the same definition of the objective function for the reduced problem because if we stay close to the maximum of the power coefficient curve, stalled conditions are avoided. This is because to produce the maximum power coefficient values, the blades operate at low angles of attack, thus staying away from the stall angle of attack.

- The lower pitch angle saturation value needs to produce a set point that still adheres to the non-linear constraint, in this case, the out-of-plane tip displacement. This point led to the same definition of the non-linear constraint on out-of-plane tip displacement.

The authors acknowledge that there might be other ways to define reduced optimization so that there is a guarantee that the blades do not incur stalling. For instance, one can use other aerodynamic quantities to express the "stall avoidance" condition. This point was clarified in the revised manuscript in the following lines:

**Revised portion:**

- **Lines 504 - 511:**
  " ...A key motivation for deriving the lower pitch saturation limit from the "reduced" optimisation in Eq. (18) is to systematically obtain minimum pitch schedules that comply with the constraints imposed in COFLEXOpt optimised operating points and avoid stall. By defining an objective function that maximises aerodynamic efficiency (i.e. the power coefficient) and retaining the OoP tip displacement constraint, we ensure that at full load, the minimal-pitch operating point (for any rotor speed–wind speed combination) remains above the stall onset value. This preserves aerodynamic stability and avoids stalled blades even if the turbine briefly operates at that minimal pitch. In contrast, simpler schedules (e.g., setting $j(\hat{V})$ to the pitch angle at rated conditions) may produce stalled conditions or violate tip-displacement limits for wind speeds in full-load operations. ... "

**Technical Corrections**

1. Line 165: The mention of "direct drive" appears misplaced and should be revised for clarity.

   **Response:**

Thank you for this feedback. As we only consider the rotor rotational speed throughout the paper, and for the sake of clarity, we do not need to mention the generator rotational speed here. Hence, we decided to leave out this line.

2. Equation 2 needs to be revised.

**Response:**
Thanks for noticing the error in Equation 2. We revised it as follows.

**Revised Portion:**

- **Equation 2:**
  " . . .

$$C_Q = \frac{Q}{\frac{1}{2}\rho V^2 \pi R^{\mathbf{3}}} \, ,$$

  . . ."

3. It is "HAWCStab2" instead of "HAWC2STAB".

**Response:**
Thanks for noticing the error in the caption of Fig. 17. We revised it as follows.

**Revised portion:**

- **Fig. 17 - Caption:**
  " Comparison of steady states (**dots**) calculated from the time-domain HAWC2 simulation and prescribed operating points (**lines**) from COFLEXOpt based on  HAWCStab2 linearisations for the four different strategies. . . ."

**References:**

Bortolotti, P., Bottasso, C. L., Croce, A., and Sartori, L. (2019). Integration of multiple passive load mitigation technologies by automated design optimization—The case study of a medium-size onshore wind turbine. Wind Energy, 22(1), 65–79. link

Brandetti, L., Liu, Y., Mulders, S. P., Ferreira, C., Watson, S., and van Wingerden, J. W. (2022). On the ill-conditioning of the combined wind speed estimator and tip-speed ratio tracking control scheme. Journal of Physics: Conference Series, 2265(3), 032085. link

Iori, J., McWilliam, M. K., and Stolpe, M. (2022). Including the power regulation strategy in aerodynamic optimization of wind turbines for increased design freedom. Wind Energy, 25(10), 1791–1811. [link]

Mulders, S. P., Liu, Y., Spagnolo, F., Christensen, P. B., and van Wingerden, J. W. (2023). An iterative data-driven learning algorithm for calibration of the internal model in advanced wind turbine controllers. IFAC-PapersOnLine, 56, 8406–8413. [link]

Sartori, L., Bellini, F., Croce, A., and Bottasso, C. L. (2018). Preliminary design and optimization of a 20MW reference wind turbine. Journal of Physics: Conference Series, 1037, 042003. [link]

**Response to Reviewer 2**

This is a very clearly written manuscript that discusses a new control strategy that combines feedforward torque and pitch control, using optimized control commands scheduled using a wind speed estimator, with feedback control to ensure the rotor speed setpoint is tracked. The paper does a nice job of showing the importance of optimizing pitch and rotor speed setpoints in the partial load region as a function of wind speed, using an aeroelastic turbine model with rotor flexibility, rather than assuming a single optimal tip-speed ratio and blade pitch. This is especially important for highly flexible rotors where blade deformations due to thrust affect the aerodynamic properties of the rotor throughout the partial load region.

This paper builds on previous work examining how turbine models including blade flexibility can be used to optimize control set points while adhering to design constraints (and the advantages over traditional control laws). Specifically, this paper extends previous ideas by presenting a closed-loop control strategy to implement the intended set points using a wind speed-estimator-based combined feedforward/feedback controller with smooth setpoint switching between the partial load and full load regions. The incorporation of blade tip displacement constraints in the set point optimisations is another important contribution.

I don't have any major concerns with the paper, but there are several areas where I believe corrections or clarifications are needed or some additional analyses would help provide more value.

**Response:**
Thank you for your feedback and recognition of our contributions. In the following section, we will provide corrections and clarifications to the points you raised.

**Specific Comments**

1. Pg. 4, ln. 91:
   Given the similarity of this work to Pusch et al. 2023, please explain the differences between that study and the research in this paper here.

   **Response:**
   Thank you for raising this point. We improved the description of the general NLP that is solved by COFLEXOpt in the revised manuscript in Section 4.1 in response to a similar point raised by the other reviewer. The contributions provided in the introduction describe the novelties (contributions) we bring with this work with respect to the current state-of-the-art described in the literature. To further highlight the differences with Pusch et al. 2023. Moreover, we revised the contribution point as follows.

**Revised portion:**

- **Lines 96 - 98:**

  " . . . Providing a set point optimisation scheme called COFLEXOpt calculating set points over the complete turbine operating range **using one optimisation problem**, adhering to operational and structural load constraints, and without the need for explicit definition of the partial to full load transition point; . . . "

2. Pg. 8, ln. 207:
   Can you add the resolution of the rotor speed, wind speed, and pitch angles that make up the 27,000 points?
   Also, can you clarify if wind shear is included in the inflow?

   **Response:**
   Thank you for your suggestion. A description of the nonconstant resolution of the three-dimensional power coefficient map is now added to the paper. No wind shear was included in the calculation of steady-state performance as HAWCStab2 only operates with uniform, constant inflow.

   **Revised portion:**

   - **Lines 216 - 225:**
     " . . . To balance computational effort and accuracy, the spacing in our grid is variable: it is refined in regions of particular interest—such as near the rated wind speed, where loads have a pronounced effect—and coarser in less critical regions. We then use HAWCStab2 to obtain the steady-state coefficients over a three-dimensional grid with 27 thousand operating points spanning various combinations of rotational speeds, wind speeds, and pitch angles. Specifically, the grid consists of:

     - 20 rotor speeds $\omega$ (from 2 to 4 $\mathrm{min}^{-1}$ in 1 $\mathrm{min}^{-1}$ steps, from 5 to 9.5 $\mathrm{min}^{-1}$ in 0.5 $\mathrm{min}^{-1}$ increments, and from 10 to 16 $\mathrm{min}^{-1}$ in 1 $\mathrm{min}^{-1}$ steps),
     - 30 wind speeds $V$ (from 2 to 7 $\mathrm{m\ s}^{-1}$ in 1 $\mathrm{m\ s}^{-1}$ steps, from 8 to 12.5 $\mathrm{m\ s}^{-1}$ in 0.5 $\mathrm{m\ s}^{-1}$ increments, and from 13 to 26 $\mathrm{m\ s}^{-1}$ in 1 $\mathrm{m\ s}^{-1}$ steps).
     - 45 pitch angles $\beta$ (from $-5$ deg to 4.5 deg in 0.5 deg increments and from 6 deg to 30 deg in 1 deg increments).

No wind shear is considered here—i.e., we assume a spatially uniform inflow. This uniform inflow assumption arises from a limitation of HAWCStab2. In principle, it would be possible to incorporate wind shear by generating performance tables with a time-domain-based simulation tool such as HAWC2. However, creating such a large number of required operating points would be computationally infeasible.
. . . "

3. Pg. 10, ln. 241: "These likely unrealistic large torsional deformations. . ."
Please explain why you believe these are unrealistic deformations..

**Response:**
Thank you for raising this point. We added a brief explanation of why these deformations are very unlikely to happen in steady-state operating conditions in real-world scenarios.

**Revised portion:**

- **Lines 261 - 265:**

  " . . . Under such conditions, large torsional deflections occur and, in turn, degrade performance while reducing loads. However, these operating points, corresponding to rotational speeds above $9 \ \mathrm{min}^{-1}$ and wind speeds above $13 \ \mathrm{m\,s}^{-1}$, lie well outside the normal steady-state operating conditions of the IEA 15 MW RWT. Consequently, these extreme deformations are not expected during typical turbine operation and are therefore considered unrealistic. . . . "

4. Section 4, 1st paragraph: Minor point.
It would be nice to mention what Section 4.2 covers in this introduction.

**Response:**
Thanks for this feedback. A mention of Section 4.2 has been added here.

**Revised portion:**

- **Lines 278 - 283:**

  " This section introduces the COFLEXOpt set point optimiser, which determines optimal operational points for large, flexible wind turbines. In Sect. 4.1, we formulate the optimisation problem for selecting set points based on turbine performance metrics and then explain the structure and implementation of the solver. Then, in Sect. 4.2 we show an illustrative example of the solution of the optimisation problem for two different wind speeds. Finally, in

Sect. 4.3, we carry out set point optimisation for different control strategies. "

5. Page 11, ln. 262: "As a consequence, the rated wind speed and operating regions were predefined."
This doesn't appear to be true. In Pusch et al. 2023, Section 3.1 states that "rated generator torque and speed are not pre-defined herein and are subject to optimisation as well... the ratio of rated generator torque and speed is determined at the smallest wind speed where a given value of rated generator power is reached."
Can you clarify in more detail how your approach differs from this previous study?

**Response:**
Thank you for raising this point. To avoid confusion with definitions of variables used in their work, in this response, we will use Pusch et al. 2023 notations. First, as shown in Tables 2, 3 and 4 from Pusch et al. 2023, their approach changes the objective function and constraints based on the "detected" control region, while in our approach, the objective function and constraints remain the same for the entire operating range of the wind turbine. This difference has the primary effect that the set point optimisation approach of Pusch et al. set point optimiser needs an additional sub-routine to change the NLP definition in full load (above rated). We acknowledge that, in some cases, the two approaches can lead to the same results.
However, using a different NLP definition in full load has the following limitation:

- Once the full load (above rated) region is detected $\overline{V} > V^{\mathrm{rtd}}$ and the NLP definition has been changed, the set point optimiser cannot freely choose alternative objectives and constraints to be satisfied. In fact, when applying the additional constraint $\tau = \tau^{\mathrm{rtd}}$, the problem reduces to finding the collective pitch angle $\beta^*$ that satisfies the operating points defined by the values $(\omega^{\mathrm{rtd}}, \overline{V}, \beta_{\mathrm{col}}, \tau^{\mathrm{rtd}})$. Notice that at this point, in above rated, the optimisation problem reduces to solving a non-linear equation $\overline{P}(\beta_{\mathrm{col}}) = P^{\mathrm{rtd}}$. Applying de-rating techniques $(P = \gamma P^{\mathrm{rtd}}$ with $\gamma < 1$ ) or additional constraints that are of interest for the wind energy community, for instance, a limit on the relative velocities encountered by blade sections $V_{\mathrm{rel}} < V_{\mathrm{rel,\ limit}}$ would require a complete re-shaping of the switching sub-routines.

Since the objective functions and constraints remain the same for the entire operating range of the wind turbine, our approach allows the user to account for different objectives, being agnostic to the operating region, and seamlessly integrate the set-point optimiser within the controller that we defined, with the possibility of extending it to perform online set point optimisation. We added a brief clarification of this in the following revised section.

**Revised portion:**

- **Section 4.1:**

  **See the Revised portion in response to Reviewer 1 - Specific Comments - 2. Objective Function**.

6. Pg. 13, ln. 310: "due to the small contribution given by the torque coefficient term"
   You explain that the weighting term $w_1$ for the torque coefficient should be small, but how did you choose the specific value?
   What value was finally used?

   **Response:**
   Thank you for your feedback. We clarified how this weighting term was chosen and added the final value used in this work, as seen in the following revised lines.

   **Revised portion:**

   - **Lines 329 - 332:**

     " ... In the remainder of this work, we set $w_1 = 0.01$. Because the power coefficient surface is relatively flat around its maximum in partial-load conditions, this small weighting factor has a negligible impact on the optimal set points in that region. However, it is sufficient to ensure unique solutions in the full-load region by regularising the objective function. ... "

7. Pg. 16, ln. 343: "where the blades pitch in to relieve thrust force"
   Should this be pitch "out"? Larger blade angles (from pitching out) would lead to lower thrust generally.

   **Response:**
   Thank you for pointing out this inconsistency. To avoid any confusion stemming from the terms "pitching in" and "pitching out," we have revised the manuscript to use "pitch to stall" and "pitch to feather." This revision ensures a clearer terminology.

   **Revised portion:**

   - **Lines 390 - 392:**

     " ... The optimisation framework allows pitching to stall, counteracting the effects of structural torsion on the blade and increasing the power output in the partial load region, as shown in the generator power plot. A different trend

is observed in the constrained strategies, where the blades pitch to feather to relieve thrust force and facilitate the decrease in OoP tip displacement. ... "

8. Eq. 10:
I believe the inertia term "J" should be in the denominator of both of the fractions on the right hand side of the first line.

**Response:**
Thank you for noticing the error in Eq. (10). We have updated it with the suggested correction.

**Revised portion:**

- **Equation 10:**

  " ...

$$\begin{cases} \dot{\hat{\omega}} = \dfrac{\rho \hat{V}^3 \pi R^2 C_P(\omega, \hat{V}, \beta)}{2\underline{J}\omega} - \dfrac{K_{\mathrm{g}}}{J} Q_{\mathrm{g}}\,, \\ e_{\hat{\omega}} = \omega - \hat{\omega}\,, \\ \hat{V} = K_{\mathrm{W,P}} e_{\hat{\omega}} + K_{\mathrm{W,I}} \int e_{\hat{\omega}}(\tau) d\tau\,, \end{cases}$$

  ... "

9. Pg. 19, ln. 399: "able to estimate the wind speed at a steady state with a significantly smaller error"
Can you discuss what might cause the small error in the wind speed estimates for the Flex. 2 case?
Are there additional degrees of freedom in the simulation that aren't in the wind speed estimator model?

**Response:**
Thank you for noticing this discrepancy. We double-checked the results of the simulations that we performed to test the wind speed estimators and found an issue in calculating the steady-state error. We fixed the calculation and modified ln. 399, and the related figure and table. An even smaller discrepancy that still remains may be due to the flexibility of the tower, which cannot be modelled by the linearised solver HAWCStab2 (that produces the $C_P$ table for the WSEs) and is active as a DoF in HAWC2 simulations.

**Revised portions:**

- **Table 3:**

| WSE Case | Model | $C_P$ table | $\max\left(|e_{\hat{V}}|\right)$ at steady state |
|----------|-------|-------------|-------------------------------------------------|
| **Rigid** | Rigid | $C_P(\lambda,\beta)|_{V=9 \text{ m s}^{-1}}$ | 3.5% |
| **Flex. 1** | Flexible | $C_P(\lambda,\beta)|_{V=9 \text{ m s}^{-1}}$ |  2.5% |
| **Flex. 2** | Flexible | $C_P(\omega,V,\beta)$ |  0.5% |

- **Figure 10:**

[Figure]

**Figure 10.** Evaluation of wind speed estimation accuracy with different WSE configurations. Percentage error in estimated wind speed ($e_{\hat{V}}$) as a function of actual wind speed during a simulation with uniform wind steps ranging from 3 to 11 metres per second. Data points represent the average of the final 100 seconds of each wind step after reaching steady state. The *Flex. 2* results, obtained using HAWC2 simulations for the IEA 15 MW RWT, demonstrate the improved accuracy of using the three-dimensional $C_P(\omega,\hat{V},\beta)$ table to reduce estimation errors in the partial load region.

10. Pg. 21, ln. 423:
    To match the description of the feedback pitch command in this sentence, you could state that when $e_\omega > 0$, $\Delta Q_{\text{g, FB}}$ should similarly be negative to accelerate rotor speed.

    **Response:**
    Thanks for your feedback. We added the suggested statement to help clarify the sign of the feedback term.

    **Revised portion:**

    - **Lines 479 - 481:**

        " ... where the two gains for the generator torque contribution $K_{\text{P,Q}}$ and $K_{\text{I,Q}}$ must be defined so that $\underline{\Delta Q_{\text{g,FB}} < 0 \text{ leads to acceleration}}$ of the rotor rotational speed when $e_\omega > 0$, ... "

11. Eq. 17:

    Is $\beta_{\max}$ also 30 degrees in this case?

    **Response:**

    Thank you for raising this point. Yes, the value we used for "normalisation" in this formula was the same as in the previous sections. We added a comment clarifying this in the revised manuscript.

    **Revised portion:**

    - **Lines 496 - 499:**

      " ...where $Q_{\mathrm{g,\,rated}}$ and $\beta_{\max}$ represent the upper saturation limits of the generator torque and collective pitch angle, respectively. In contrast, the function $j(\hat{V})$ represents the lower varying saturation limit for the collective pitch angle. We developed a new methodology to obtain $j(\hat{V})$. ... "

12. Pg. 23, ln. 447: "producing stable operating points for the wind turbine in the full load region"

    Please explain how this choice of $j(V)$ produces operating points that are "stable" and how this differs from the strategy used by Abbas et al. 2022.

    How would the stability compare to other simpler choices of $j(V)$, such as setting it equal to the pitch angle at the rated wind speed?

    **Response:**

    Thank you for raising this point. First, as already noticed in the answer to the Reviewer 1 on this point, the strategy used by Abbas et al. 2022 seems to lead to similar results to what we obtained (see their Fig. 9 (a), orange line and our Fig. 13, blue and orange dashed lines). Nonetheless, in their work, it is unclear how these schedules were obtained. When we mention that these points are stable, we mean that in case of a transient situation in which the turbine finds itself operating at $\left(\omega^*, \overline{V}, \beta_{\min}(\overline{V})\right)$ the blades would still operate in non-stalled conditions, thus avoiding aerodynamic instabilities due to stall. Moreover, thanks to how we defined the reduced optimisation of Eq. (18), operating at the minimum pitch schedule would still lead to compliance with the non-linear constraints. This would not be true if $j(V)$ could be set to the pitch angle at the rated wind speed. To clarify this point we revised Sect. 5.3 (see answer to first reviewer).

    **Revised portion:**

    - **See the Revised portion in response to Reviewer 1 - Specific Comments - 5. Saturation Schedules**.

13. Pg. 23, ln. 455: "In the partial load region, $\Delta\omega_{\text{bias}_2} = 0$ and $\Delta\omega_{\text{bias}} < 0$"
    Is there a sign error somewhere in Eq. 16 or 19?
    Otherwise I think $\Delta\omega_{\text{bias}}$ would only be negative in the partial load region if the gain $K_{\text{bias}_1}$ is negative (I'm assuming you intend for the gains to be positive values).

    **Response:**
    Thanks for noticing this inconsistency. Eq. (16) has indeed an error. We revised it so that it would lead to positive values when used in the set-point smoother.

    **Revised portion:**
    - **Eq. (16):**

      "

      $$\Delta\omega_{\text{bias}_1} = \frac{Q_{\text{g, rated}} - Q_{\text{g}}}{Q_{\text{g, rated}}},$$

      . . .

      "

14. Pg. 24, ln. 464:
    How did you design this low-pass filter? What cut-off frequency was used?

    **Response:**
    Thank you for raising this point. We employ a discrete-time version of a first-order low-pass filter to remove high-frequency components from the $\Delta\omega_{\text{bias}}$ signal. Because we want the set-point smoothing technique to behave similarly to that described for the IEA 15 MW RWT in Abbas et al. (2022), we use the same cut-off frequency ($0.2\pi$ rad/s) specified in the publicly available repository of the IEA 15 MW RWT (Servodyn input file '*IEA-15-240-RWT-Monopile_DISCON.IN*') We have clarified this detail in the revised manuscript accordingly.

    **Revised portion:**
    - **Lines 521 - 522:**

      " . . . The signal $\Delta\omega_{\text{bias}}$ is also low-pass filtered to prevent high-frequency oscillations. In particular, we used a discrete-time first-order filter with a cut-off frequency of $0.2\pi \ \mathrm{rad\,s^{-1}}$. . . . "

15. Fig. 15:
    Can you explain why there is an underestimation bias in the estimated wind speed?

**Response:**
Thank you for this feedback. Our controller relies on a torque-balance wind speed estimator (WSE), which uses a look-up table of the power coefficient calculated at steady state. Even though this estimator performs well on average (and shows very low errors at steady states), we observed some discrepancies between the estimated values and the true ones in our time-domain simulations. We believe that this discrepancy arises from several factors:

- the time-domain simulations performed in HAWC2 include dynamic effects and DoFs that the rather simple single DoF aerodynamic model in the WSE does not capture;
- the dynamic performance of the WSE depends on the combined tuning/calibration of the estimator and controllers and the set-point smoothing technique;
- noise in the signals that are input to the WSE.

To improve the readability and clarify the scope of Fig. 15, we modified the WSE and controller tuning and re-ran the simulation for Fig. 15. The fixes include:

- Re-tuning of the WSE and controllers;
- Low-pass filtering of the feedforward control signals $Q^*_{\mathrm{g,\,FF}}(\hat{V})$ and $\beta^*_{\mathrm{FF}}(\hat{V})$;
- Correcting a post-processing issue in Fig. 15(d) that had introduced the wrong scaling for $e_\omega$, $e'_\omega$, and $\Delta\omega_{\mathrm{bias}}$.

We have revised the figure and its discussion accordingly in the updated manuscript.

**Revised portions:**

- **Lines 558 - 585:**

  " ... To observe this transition in detail, we have extracted a 40-second segment from a 1000-second simulation carried out with a turbulent wind field and wind shear, capturing the moment when the rotor's average wind speed crosses the rated wind speed. Figure 15 (a) compares the rotor-average wind speed (light grey) with its corresponding estimate (dark grey). Overall, the two signals align well, though the estimated value shows some high-frequency oscillations that likely stem from noise in the WSE input signals and the calibration of the WSE. Brief discrepancies also occur (e.g. near $t \approx 510\,\mathrm{s}$), which may be attributed to dynamic effects or degrees of freedom not captured by the internal model used in the WSE. To prevent the high-frequency oscillations from directly exciting the actuators, we apply a first-order low-pass filter with a cut-off frequency of $0.5\pi\ \mathrm{rad\,s^{-1}}$ to the feedforward inputs. Figures 15 (b) and (c) show, respectively, the feedforward pitch and torque commands scheduled on the true rotor-average wind speed (light grey), on the estimated wind speed (dark grey), and the actual controller outputs (green). Up to $t \approx 505\,\mathrm{s}$, the turbine remains in partial-load operation: The collective

pitch angle closely follows the feedforward command, which in turn tracks the ideal feedforward value reasonably well. Near $t = 505\,\mathrm{s}$, the generator torque saturates (Fig. 15 (c)) to maintain rated power. At that moment, the estimated wind speed in Fig. 15 (a) reaches around $10.7\,\mathrm{m\,s^{-1}}$, matching the expected rated condition. Figure 15 (d) illustrates how the set-point bias $\Delta\omega_{\mathrm{bias}}$ (blue) ensures a smooth transition from torque to pitch control. Before $t \approx 505\,\mathrm{s}$, the bias is negative, keeping the collective pitch angle saturated at its lower limit and allowing the torque controller to be active. As the system approaches rated, the bias crosses zero and effectively drives the generator torque into saturation, activating the collective-pitch controller. This gradual shift avoids abrupt changes in control action and demonstrates that the combined feedforward-feedback strategy can successfully handle transitions to full-load operation, even under turbulent inflow. Finally, while the overall dynamic performance is satisfactory, further gain scheduling or fine-tuning of the WSE and PI loops could improve transient behaviour and reduce any remaining high-frequency pitch or torque activity.... "

- **Figure 15**

[Figure]

**Figure 15.** Quantities extracted from a time domain simulation of the IEA 15 MW RWT with turbulent wind and wind shear, performed in HAWC2 with the implementation of the novel control scheme, showing the behaviour of control inputs and set-point smoothing technique values near the transition from partial load to full load. The vertical dashed line at $t \approx 505$ s marks the transition from generator torque control to collective pitch control in the full-load region. (a) Rotor-average wind speed (light grey) and estimated wind speed (dark grey). (b) Ideal feedforward collective pitch angle scheduled on the actual rotor average wind speed (light grey), feedforward scheduled on the estimated wind speed (dark grey), and the controller pitch command (green). (c) Ideal feedforward generator torque scheduled on the actual rotor average wind speed (light

grey), feedforward input scheduled on the estimated wind speed (dark grey), and the actual generator torque command (green). (d) Rotational-speed error $e_\omega$ (black), biased error $e'_\omega$ (red), and the set-point smoothing technique bias $\Delta\omega_{\mathrm{bias}}$ (blue).

16. Fig. 15b:
There is a considerable high-frequency component in the blade pitch feedforward signal (and the torque setpoint too) stemming from the high frequencies in the estimated wind speed. Can you please discuss where this comes from, and is it problematic?
The oscillations in the pitch angle could potentially increase damage to the pitch actuators. What improvements could be made to reduce the high-frequency component of the estimated wind speed?

**Response:**
Thank you for the feedback. We acknowledge that the high-frequency oscillations in the pitch and torque feedforward signals largely stem from the WSE. Because the WSE's internal model does not capture all of the degrees of freedom present in the HAWC2 simulations, the estimator can misinterpret unmodeled high-frequency oscillations with actual changes in the wind speed. When used for feedforward control, these high-frequency signals might lead to pitch and torque actuator wear. As an initial measure, we have added filters on the feedforward signals to remove most of the high-frequency components. For Fig. 15, we re-ran the simulations with these additional filters, and we observed a significant reduction in pitch high-frequency oscillations. We do note that simply filtering the feedforward signals does not entirely resolve the underlying issue. In future work, we plan to: refine the WSE model, analyse the WSE bandwidth and improve the overall control tuning.
These steps will aim to understand the root causes of high-frequency content in the WSE. However, for this study, the introduced feedforward filtering has proven sufficient to reduce the oscillations, as shown in the revised simulation results.

**Revised portion:**

- **See answer to 15.**

17. Fig. 15d:
$e_\omega$ is consistently negative across the 20 seconds of the simulation. This suggests that the combined pitch and torque control strategy is regulating rotor speed poorly. Can you discuss this?
I'm also surprised that the difference between $e_\omega$ and $e'_\omega$ is so small. Given that the difference is tiny compared to the magnitude of the rotor speed error, how does this meaningfully impact the setpoint switching?

**Response:**
Thank you for noticing the inconsistency. The authors double-checked the postprocessing of the data producing Figure 15, where we noticed an error in the calculations of the quantities shown in subplot (d). After revisiting Figure 15 and the post-processing script, the three lines shown in subplot (d) show the correct values.

**Revised portion:**

- **See answer to 15.**

18. Fig. 15:
Minor point, but in the first sentence of the caption it might be clearer to describe the variables in the order they're shown in the subplots.

**Response:**
Thank you for this feedback. The caption of Figure 15 has been revised as shown in the previous answer.

**Revised portion:**

- **See answer to 15.**

19. Pg. 28, ln. 527:
"a discrepancy in the steady-state blade deflection calculation for HAWC2 and HAWCStab2": Could you simply use HAWC2 for the steady-state calculations?

**Response:**
Thank you for this feedback. In line 527, we are comparing the steady-states of a time-domain simulation with wind steps performed with the new controller, with the prescribed set-points (obtained using COFLEXOpt), which are optimised based on steady-states quantities coming from HAWCStab2. The rationale behind the choice of HAWCStab2 (linearised aeroelastic solver) for the steady-states to be used in the set-point optimiser lies in its computation time. To evaluate the same steady-states with a time-domain-based solver (like HAWC2) would require considerably higher computation time. In cases of using the set-point optimiser inside an outer optimisation loop (for example, for co-design), it would result in extremely high computation times. We clarified this in Pg. 5, ln. 126 - ln. 128

**Revised portion:**

- **Lines 126 - 129:**

    " . . . Secondly, it provides a very fast computational time, which is crucial for evaluating performance across thousands of operating points that result from the combination of the three independent variables: rotational speed,

wind speed and collective pitch angle, with sufficiently fine resolution. Hence, this tool offers a good trade-off between calculation accuracy and computational cost for operating-point evaluations. ... "

20. Section 6.2:
It would strengthen the results of the paper to compare the controller performance in turbulent wind conditions to the performance of the simpler reference controller. For example, although the steady state results show improvements compared to the reference controller, how do the power and tip displacement compare in more realistic turbulent conditions between the novel controller design and the reference controller?

**Response:**
Thank you for this feedback. We complemented the discussion with an additional figure showing the differences in median values of tip displacement and generator power (normalised w.r.t. to the reference) in turbulent conditions for the four strategies. We also added a brief comment as follows.

**Revised portions:**

- **Lines 691 - 700:**

" ... Figure (20) compares the median values of OoP tip displacement (top panel) and generator power (bottom panel), both normalised by the reference strategy, for the new strategies across wind speeds from 5 to 15 metres per second. For *Case 1* and *Case 2*, we observe that the generator power increases by approximately five percentage points relative to the reference at the expense of higher tip displacements in the partial load region. In particular, *Case 1* shows OoP tip displacements as much as 30% above the reference at rated wind speed, which aligns with the prescribed operating points. In *Case 2*, the displacement constraint is active around $10 \mathrm{~m\,s}^{-1}$, as indicated by the orange bars converging toward unity in the top panel near $11 \mathrm{~m\,s}^{-1}$. *Case 3* follows a similar pattern at lower wind speeds (below $8 \mathrm{~m\,s}^{-1}$), but the tighter constraint on tip displacement results in values around 25% below the reference near the rated wind speed, and a corresponding lower power output in that range. All three cases behave similarly to the reference controller in full-load operations. Overall, these trends confirm that the set points derived via COFLEXOpt can be effectively tracked in turbulent inflow scenarios. "

- **Figure 20**

[Figure]

**Figure 20.** Median OoP tip displacement (*top*) and generator power (*bottom*), both normalised by the values obtained with the reference strategy for each wind speed bin across wind speeds of 5 to 15 $\mathrm{m\,s^{-1}}$. Bars represent 10-second median values obtained from six 600-second HAWC2 simulations under realistic turbulence, grouped in 1 $\mathrm{m\,s^{-1}}$ bins. The reference strategy values (unity) are shown in grey, while *Case 1* (blue), *Case 2* (orange), and *Case 3* (yellow) bars represent the values obtained with the new strategies. In *Cases 1* and *2*, power increases relative to the reference, but tip displacements rise by up to 30% in partial-load operation. *Case 3* exhibits a 25% reduction in tip displacement near rated wind speed, associated with generally lower generator power.

21. Section 6.2:
    Please mention what amount of wind shear was included in the turbulent simulations.

    **Response:**
    Thank you for this feedback. We included a mention of the wind shear characteristics that were used in the simulations.

    **Revised portion:**

    - **Lines 646 - 647:**

      " . . . Additionally, a power-law vertical wind shear was applied with an exponent of 0.2. . . . "

22. Pg. 28, ln. 555: "This consistent, positive bias. . ."

Could the presence of wind shear cause this positive bias?
For example, shear might lead to higher turbine power than predicted by a simple rotor average of wind speed used for comparison.

**Response:**

Thank you for highlighting this issue. Our wind speed estimator relies on a torque-balance approach, which uses the balance between measured generator torque and estimated rotor torque to derive the wind speed. The rotor torque is a non-linear function of wind speed; recall the following equation, which is used in the WSE.

$$\hat{Q}_{\mathrm{r}} = \frac{\rho \hat{V}^3 \pi R^2 C_P(\omega, \hat{V}, \beta)}{2J\omega}$$

For the calculation of the actual rotor average wind speed $V$, it is important to realize that different sections of the blade do not contribute equally to the overall rotor torque. This means that, under wind shear and turbulent conditions, the wind speed that contributes to producing the rotor torque is, in general, different from the rotor average wind speed.

When using a global estimate of the rotor torque, our WSE is averaging $(V^3)$ over the rotor area, exceeding $(\overline{V})^3$, causing the estimator to interpret the rotor as if it were experiencing a higher uniform wind speed.

These considerations lead to the conclusion that when wind speed is not uniform over the rotor disk, our WSE estimates a quantity that is slightly different from the rotor average wind speed that we use as a reference. However, this bias does not compromise the performance of the controller. In practice, the control scheme should track set points based on this *effective* wind speed, and COFLEX set-point mappings and feedforward inputs depend precisely on that quantity (estimated by the WSE). We also note that the small discrepancies between median values and prescribed set points likely arise because we binned results by rotor average wind speed. Although further studies could use the rotor effective wind speed output from HAWC2 (noting that its definition requires particular care, as stated in the manual), such an investigation lies beyond the scope of this work. We revised the new manuscript accordingly.

**Revised portion:**

- **Lines 662 - 671:**

  " . . . This consistent, positive bias was not observed in previous analyses and is likely driven by local wind speed fluctuations due to wind shear and turbulence. Our wind speed estimator uses a torque-balance approach, matching the measured generator torque to an estimated rotor torque, recalling the system of Eq. (10). Under wind shear and turbulence, the contribution of

blade sections to the total torque depends on the local velocities. Hence, the effective wind speed which produces the rotor torque differs from the arithmetic mean across the rotor disk. As a result, the WSE estimates an effective wind speed that differs from the rotor average wind speed, which is used as a reference here. However, this bias does not degrade the performance of the controller. In a practical scenario, the controller must adapt to this effective wind speed; the control scheme of COFLEX still holds, as our set-point mappings and feedforward inputs rely on precisely this torque-based wind speed estimate.
. . . "

- **Lines 673 - 675:**

" . . . These differences can be largely attributed to the bias between the estimated wind speed and the rotor average wind speed resulting from the simulator used for binning. This directly impacts the feedforward component in the control loop, especially at low wind speeds.
. . . "

- **Lines 723 - 726:**

" . . . Despite a slight wind speed estimation bias, which may be attributed to the difference in the estimated effective wind speed and rotor average wind speed, the controller maintained tracking of rotor speed, generator torque, and collective pitch angle under turbulent conditions.
. . . "

23. Pg. 29, ln. 563: "the expected constraint on the median value of the OoP tip displacement is satisfied with a deviation of less than 1%".
For OoP tip displacement, I would think the maximum value would be more important than the median within a wind speed bin (since even one tower strike would cause damage). Is the median value a relevant way to judge the tip displacement here?

**Response:**
Thank you for this valuable observation. We agree that, from a tower-strike perspective, the maximum out-of-plane tip displacement is of critical importance. However, the maximum displacement is primarily driven by transient effects (e.g., controller tuning and dynamic wind conditions). Consequently, imposing a hard limit on the maximum displacement would require a different control architecture (e.g., advanced IPC or MPC) that can explicitly predict and mitigate such extremes—beyond the scope of our current work.

Nonetheless, to address the probability of exceeding a safe maximum value, one could extend our steady-state constraint by incorporating a precomputed variance around the median displacement. Our median-based constraint in COFLEXOpt could be augmented to ensure that the maximum displacement remains within an acceptable safety margin. We have added a brief comment on this in the revised manuscript.

**Revised portion:**

- **Lines 679 - 686:**

  " . . . While constraining the steady-state OoP tip displacement helps reduce average deflection levels, more advanced control techniques remain necessary to mitigate the transient effects that drive the maximum values—and thus the tower-strike risk. Consequently, imposing a strict limit on the maximum displacement would require a different control approach, such as online set-point optimisation (Petrović and Bottasso, 2017) or advanced individual pitch control (Liu et al., 2022), which can explicitly predict and counteract such extremes. Nonetheless, to address the safety margin in a stochastic way, one could modify the constraint in COFLEXOpt by incorporating a precomputed variance around the median displacement. This would allow designers to ensure, a priori, that the probability of exceeding the maximum allowable OoP tip displacement remains within an acceptable margin.
  . . . "

24. Section 7:
    This is impressive work, but it would be interesting to briefly summarize ideas for improvements to the control strategy for future work. For example, could the controller be designed to better handle different amounts of wind shear?
    How could the controller be combined with IPC to better reduce maximum tip displacement?
    Could the wind speed estimator be improved to reduce the high-frequency ripple in the estimates?
    Are there ideas for reducing the bias in the wind speed estimator?

    **Response:**
    Thank you for this feedback. We have briefly summarised the capabilities of online set-point optimisation (which is a natural development of COFLEX) for potential improvements and future work in the conclusions. Because the conclusions are already fairly long, we kept this addition concise—and have instead addressed specific improvements (for example, related to the WSE) in the paragraphs where they are most directly relevant.

**Revised portion:**

- **Lines 734 - 736:**

[revised manuscript text omitted]